# SUSA2 is an F-box protein required for autoimmunity mediated by paired NLRs SOC3-CHS1 and SOC3-TN2

Wanwan Liang[1,2], Meixuezi Tong[1,2] & Xin Li [1,2✉]

Both higher plants and mammals rely on nucleotide-binding leucine-rich repeat (NLR) immune receptors to detect pathogens and initiate immunity. Upon effector recognition, plant NLRs oligomerize for defense activation, the mechanism of which is poorly understood. We previously showed that disruption of the E3 ligase, Senescence-Associated E3 Ubiquitin Ligase 1 (SAUL1) leads to the activation of the NLR SOC3. Here, we report the identification of *suppressor of saul1 2* (*susa2*) and *susa3* from the *saul1-1* suppressor screen. Pairwise interaction analysis suggests that both SUSA proteins interact with components of an SCF[SUSA2] E3 ligase complex as well as CHS1 or TN2, truncated NLRs that pair with SOC3. *susa2-2* only suppresses the autoimmunity mediated by either CHS1 or TN2, suggesting its specific involvement in SOC3-mediated immunity. In summary, our study indicates links between plant NLRs and an SCF complex that may enable ubiquitination and degradation of unknown downstream components to activate defense.

[1] Michael Smith Laboratories, University of British Columbia, Vancouver, BC V6T 1Z4, Canada. [2] Department of Botany, University of British Columbia, Vancouver, BC V6T 1Z4, Canada. ✉email: xinli@msl.ubc.ca

Plants have evolved two types of immune receptors to recognize pathogens and turn on immune responses[1,2]. Plasma membrane-localized pattern recognition receptors (PRRs) recognize pathogen or microbe-associated molecular patterns (PAMPs/MAMPs) to induce PAMP-triggered immunity (PTI)[3]. Adapted pathogens can deliver diverse effectors into plant cells to interfere with PTI and enhance virulence. To counteract, plants utilize intracellular polymorphic resistance (R) proteins to perceive these effectors either through direct binding or monitoring perturbations of host proteins caused by effectors, to initiate a more robust defense response termed effector-triggered immunity (ETI). ETI often leads to hypersensitive response (HR), a programmed cell death that has been hypothesized to restrict pathogen growth at the local infection site[4–7].

Most of the studied R genes encode nucleotide-binding and leucine-rich repeat proteins (NLRs). Canonical NLRs are divided into two subgroups, with coiled-coil (CC; CC-type NLRs are CNLs) or Toll interleukin-1 receptor (TIR; TIR-type NLRs are TNLs) domain at their N termini[8]. TNLs and CNLs seem to utilize different downstream signaling components; Enhanced Disease Susceptibility 1 (EDS1) and NON-RACE SPECIFIC DISEASE RESISTANCE 1 (NDR1), respectively[9]. How NLRs are activated to transduce signals downstream remains unclear. Recent structural analysis showed that similar to the inflammasome formed during animal NLR activation, Arabidopsis CNL HopZ-ACTIVATED RESISTANCE 1 (ZAR1) oligomerizes into an active wheel-like pentameric ZAR1 resistosome with RESISTANCE RELATED KINASE 1 (RKS1; serving as an adaptor) and uridylylated AvrPphB SUSCEPTIBLE 1-LIKE 2 (PBL2)[10,11]. PBL2 acts as a decoy which can be uridylylated by *Xanthomonas campestris* effector AvrAC, and subsequently recognized by ZAR1[12]. Whether other plant NLRs similarly form such resistosomes upon activation remains to be elucidated. It is also unclear what molecular events occur upon resistosome formation to activate defense.

Both mammal and higher plant NLRs need to be correctly folded, assembled, and maintained at a signal-competent state during activation. This process involves Heat Shock Protein 90 (HSP90) chaperones, which are structurally and functionally conserved among eukaryotes, together with the two other co-chaperones Required for MLA12 Resistance 1 (RAR1) and Suppressor of the G2 Allele of SKP1 (SGT1)[13,14]. Interference with HSP90 has been reported to attenuate NLR-mediated immune responses[15,16]. The HSP90 inhibitor geldanamycin can interfere with HR response and disease resistance mediated by Resistant to *Pseudomonas syringae* 2 (RPS2) in response to *P. syringae* pv. *tomato* (*P.s.t.*) DC3000 carrying avrRpt2 effector[15]. Similarly, mutations in HSP90 also compromise immunity mediated by Resistance to *P. syringae* pv. *maculicola* 1 (RPM1)[16].

Many plant NLRs form hetero-pairs in ETI[8,17–20]. Arabidopsis TNLs Resistance to *P. syringae* 4 (RPS4) and Resistance to *Ralstonia solanacearum* 1 (RRS1) hetero-dimerize through the TIR domains to form the RPS4-RRS1 complex, where effector recognition is mediated by the integrated decoy WRKY domain at the C terminus of RRS1[21,22]. In monocot rice, two CNLs Resistance gene analog 4 (RGA4) and RGA5 cooperate as another well-characterized paired NLR receptor complex to recognize *Magnaporthe oryzae* effectors Avr-Pia and Avr1-CO39[18,19]. Furthermore, TNL protein Suppressor of *chs1-2*, 3 (SOC3) associates with its partners Chilling Sensitive 1 (CHS1) or TIR-NBS 2 (TN2) to guard E3 ligase Senescence-Associated E3 Ubiquitin Ligase 1 (SAUL1; also named PUB44, PLANT U-BOX 44) homeostasis[23–25]. In all these paired NLR cases, the involved genes are encoded by head-to-head arranged genes, presumably to facilitate co-expression of the pair.

The *chs1-2* mutant induces constitutive defense responses at low temperature (16 °C or below), which include extensive cell death, upregulation of defense-associated genes, and enhanced resistance to pathogens[26]. The knockout mutant of *saul1* displays similar autoimmune phenotypes at 21 °C, which is dependent on EDS1, Phytoalexin Deficient 4 (PAD4), and SGT1b[27,28]. In suppressor screens using either *chs1-2* or *saul1-1*, SOC3 was identified[25,29]. Genetic and biochemical evidence showed that SOC3 pairs with either CHS1 or TN2, both truncated NLRs with only TIR and NB (TN) domains, to guard the homeostasis of E3 ligase SAUL1[24,25]. How the paired immune receptor complexes SOC3–CHS1 or SOC3-TN2 are assembled, activated, and regulated during defense activation remains unclear.

Here, we report the identification of two new *suppressor of saul1-1* (*susa*) mutants isolated from the *saul1-1* suppressor screen; one with a mutation in *HSP90.3*, while the other carries a mutation in the F-box domain-containing protein SUSA2. Our biochemical and genetic analysis suggests that both SUSA proteins may participate in an SCF complex required for SOC3–CHS1 or SOC3-TN2 paired NLR immune receptor-mediated autoimmunity.

## Results

**Identification, characterization, and positional cloning of the susa2-1.** Previously, we uncovered that E3 ligase SAUL1, a positive regulator of PTI, is guarded by TNL SOC3. SOC3 pairs with a truncated TN, CHS1 or TN2, to monitor the homeostasis of SAUL1[24,25]. Knockout *saul1-1* plants exhibit seedling lethality and enhanced disease resistance that are dependent on SOC3 and CHS1, while the autoimmunity upon overexpression of *SAUL1* relies on SOC3 and TN2[24,25,30]. To search for the SAUL1 ubiquitination substrate and defense signaling components involved in SOC3-mediated immunity, we performed a *suppressor of saul1-1* (*susa*) forward genetic screen to identify genetic suppressors of *saul1-1*[25]. In addition to many *susa1/soc3* alleles[25], *susa2-1 saul1-1* and *susa3-1 saul1-1* were isolated (Fig. 1a). Both mutants largely suppress the dwarfism of *saul1-1*. When cell death was examined by trypan blue staining, *susa3-1 saul1-1* showed partial, while *susa2-1 saul1-1* showed full suppression (Fig. 1b). However, in an ion leakage assay, both *susa* mutants suppressed the high ion leakage in *saul1-1* back to Col-0 wild-type (WT) levels (Fig. 1c). Similarly, enhanced disease resistance against virulent oomycete pathogen *Hyaloperonospora arabidopsidis* (*H.a.*) Noco2 in *saul1-1* was partially or completely suppressed by *susa3-1* and *susa2-1*, respectively (Fig. 1d). Together, these data indicate that *susa2-1* is full, while *susa3-1* is a partial suppressor of *saul1-1*. When *susa2-1 saul1-1* and *susa3-1 saul1-1* were backcrossed with the *saul1-1* parent, the F1 plants resembled *saul1-1*, indicating that both suppressors are recessive. Therefore, *susa2-1* and *susa3-1* most likely carry loss-of-function mutations.

To identify the molecular lesion responsible for *susa2-1*, *susa2-1 saul1-1* (in Col-0 background) was crossed with the Landsberg *erecta* (*Ler*-0) ecotype to generate an F2 mapping population. Linkage analysis revealed that *susa2-1* is within a 1.7 Mb region between markers K19E1 and MUA2 on chromosome 5 (Fig. 2a). Whole-genome sequencing (WGS) was then carried out on genomic DNA extracted from *susa2-1 saul1-1* plants. Comparison of the *susa2-1* sequence with the Col-0 WT reference genome in the mapped region revealed three non-synonymous mutations in three genes, At5g56180, At5g56190, and At5g56890, all of which caused amino acid substitutions (Fig. 2b).

To determine which of the three mutations is responsible for the *susa2-1* phenotype, T-DNA mutants of the above-mentioned three genes, including SALK_093650 and SALK_020877 carrying insertions in the 6th and 7th intron of At5g56180, respectively, (Fig. 2c), SALK020826 with insertion in the 9th exon of At5g56190, and CS878039 carrying insertion in the 11th exon

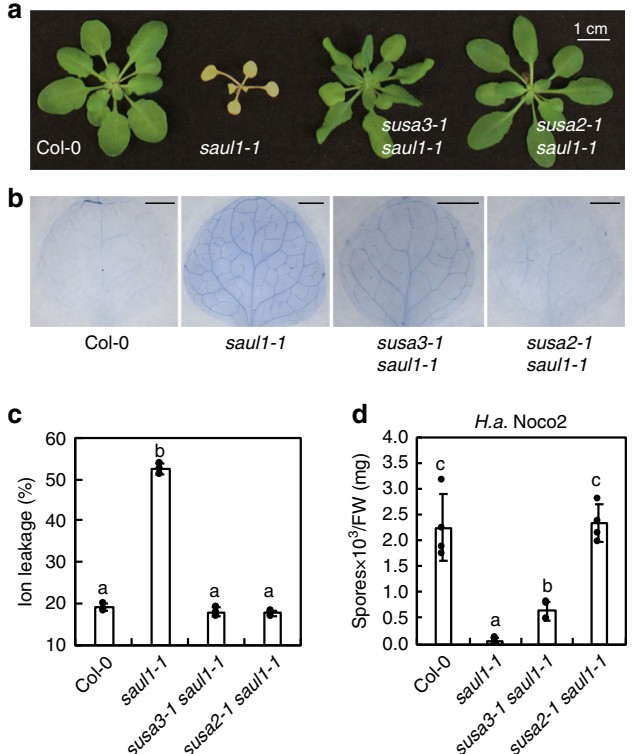

**Fig. 1 Characterization of the *susa2-1 saul1-1* and *susa3-1 saul1-1* mutants.**
**a** Morphology of 3.5-week-old Col-0, *saul1-1*, *susa3-1 saul1-1,* and *susa2-1 saul1-1* plants. Plants were grown on 1/2 MS medium for 10 days and then transplanted to soil for another two weeks before the picture was taken. **b** Trypan blue staining of 14-day-old Col-0, *saul1-1*, *susa3-1 saul1-1,* and *susa2-1 saul1-1* plants. Plants were grown on 1/2 MS medium for 7 days and then transplanted to the soil for another week to allow *saul1-1* phenotype to develop. The scale bar shows 1 mm. **c** Ion leakage measurement of the indicated plants shown in **a**. Ion leakage was calculated as the percentage of the conductivity before autoclaving over that after autoclaving. Six leaf discs were used per genotype in 20 ml $H_2O$. Error bars represent means ± SD (one-way ANOVA, SPSS Statistics, $n = 3$, $p < 0.01$). Experiments were repeated three times with similar results. **d** Quantification of oomycete pathogen *H.a.* Noco2 growth on the indicated genotypes. Two-week-old plants were evenly sprayed with *H.a.* Noco2 conidiospores at a concentration of 100,000 spores/ml water. Quantification of conidia growth on leaf surface per mg FW was determined 7 days post-inoculation (dpi). Error bars represent means ± SD (one-way ANOVA, SPSS Statistics, $n = 4$, $p < 0.01$). Experiments were repeated three times with similar results.

of At5g56890, were obtained from the *Arabidopsis* Biological Resource Center (ABRC). Homozygous T-DNA lines were crossed with *saul1-1* and double mutants were identified in the F2 population. SALK_093650/*susa2-2* and SALK_020877/*susa2-3* both suppressed *saul1-1*, but not SALK020826 and CS878039 (Fig. 2d and Supplementary Fig. 1), suggesting that *SUSA2* is likely *At5g56180*, which was also named *ARP8* (ACTIN-Related Protein 8) due to the presence of an ACTIN domain besides an F-box motif (Fig. 2e)[31]. A transgene complementation experiment was further performed to confirm the identification of *SUSA2*. When the genomic region of *At5g56180* driven by its native promoter (pCambia1305 *SUSA2::SUSA2*, hereafter *SUSA2::SUSA2*) was transformed into *susa2-1 saul1-1*, among 36 T1 transformants obtained, 23 exhibited complementation and displayed *saul1*-like phenotypes (Fig. 2f, g). We, therefore, concluded that *SUSA2* is indeed *At5g56180*.

In *saul1-1* mutant plants, the ubiquitination substrate of E3 ligase SAUL1 may accumulate. To test the possibility that SAUL1 might target SUSA2 for ubiquitination and further degradation, physical interaction was examined by co-immunoprecipitation (co-IP) using transiently expressed proteins in *Nicotiana benthamiana* (*N. benthamiana*). WT SUSA2-FLAG could fully complement *susa2-1 saul1-1* (Supplementary Fig. 2A, B), suggesting that the FLAG tag does not affect the function of SUSA2. As WT SUSA2 is of extremely low abundance and its protein band is undetectable on western blots (Supplementary Fig. 2C), we engineered a dominant-negative (DN) form of SUSA2, where the F-box domain (residues 40–86) was deleted, likely abolishing the interaction of the F-box with SKP1/ASKs in the SCF E3 ligase complex but maintaining its interaction with its substrate, preventing self-ubiquitination and degradation of itself and its substrate. Such an approach has been used successfully for studying F-box proteins previously[32,33]. As expected, the DN-SUSA2-FLAG expressed well (Supplementary Fig. 2C). As shown in Supplementary Fig. 3A, B, DN-SUSA2-FLAG could not pull down HA-SAUL1 and neither could SAUL1-C29A-FLAG (a DN form of SAUL1 disrupting the RING domain of the E3; it serves to stabilize SAUL1) pull-down SUSA2-HA in a reciprocal IP, indicating that SAUL1 does not interact with SUSA2. In addition, when DN-SUSA2-FLAG was co-expressed with SAUL1 in *N. benthamiana*, no change in DN-SUSA2 level was observed (Supplementary Fig. 3C). This was confirmed in stable transgenic *Arabidopsis* plants co-expressing the two transgenes (Supplementary Fig. 3D), confirming that SUSA2 is unlikely a ubiquitination substrate of SAUL1.

F-box proteins have been shown to be part of SCF E3 ligase complexes, which bring their ubiquitination substrate into proximity with an E2 ubiquitin-conjugating enzyme, leading to degradation of the substrate[34]. As SUSA2 is an F-box protein, it likely functions as part of an SCF E3 ligase to ubiquitinate its substrate. To examine whether SUSA2 might target SAUL1 for ubiquitination and further degradation, DN-SUSA2-FLAG was co-expressed with HA-SAUL1. As shown in Supplementary Fig. 3E, the presence of DN-SUSA2-FLAG did not stabilize SAUL1, indicating that SAUL1 is unlikely a ubiquitination substrate of SUSA2. This is consistent with our previous observation that these two proteins did not interact with each other.

**Characterization and positional cloning of *susa3-1*.** Another suppressor *susa3-1 saul1-1* partially suppresses the autoimmune phenotypes of *saul1-1* (Fig. 1). To identify the responsible mutation in *susa3-1*, *susa3-1 saul1-1* (in Col-0 background) was crossed with L*er*-0 to generate an F2 mapping population. Linkage analysis revealed that the *susa3-1* mutation also mapped to the same region as *susa2-1* on chromosome 5. As both *susa* mutants are recessive, we tested whether *susa3-1* is allelic to *susa2-1* by crossing *susa3-1 saul1-1* with *susa2-1 saul1-1*. The resulting F1 plants resembled *saul1-1* (Fig. 3a), indicating that *susa3-1* and *susa2-1* complemented each other; they should harbor mutations in different genes.

When examining this mapped region for known immune regulators, we noticed that there are *HSP90* family genes. As loss-of-function mutations in *HSP90s* were reported to attenuate NLR-mediated immunity[15,16], and NLR SOC3 is activated in *saul1-1*, there is a possibility that mutation in *HSP90s* might suppress *saul1-1*. Sanger sequencing was then performed on *susa3-1 saul1-1* plants to search for *HSP90* mutations. A C299T mutation was found in the 3rd exon of *HSP90.3*, which causes an S100F amino acid change. Intriguingly, this same S100F change was previously reported in *hsp90.3-1* and *hsp90.2-2* mutants[35–37],

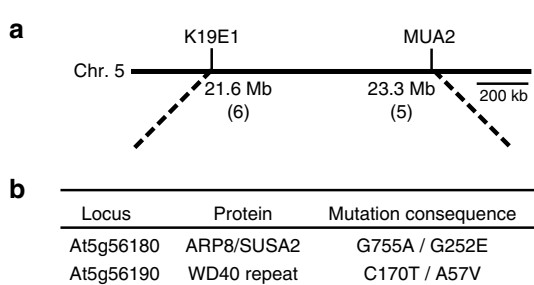

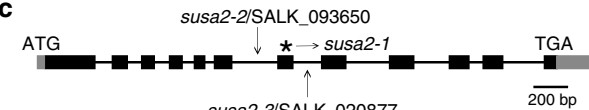

| Locus | Protein | Mutation consequence |
|---|---|---|
| At5g56180 | ARP8/SUSA2 | G755A / G252E |
| At5g56190 | WD40 repeat | C170T / A57V |
| At5g56890 | kinase | G2683A / A895T |

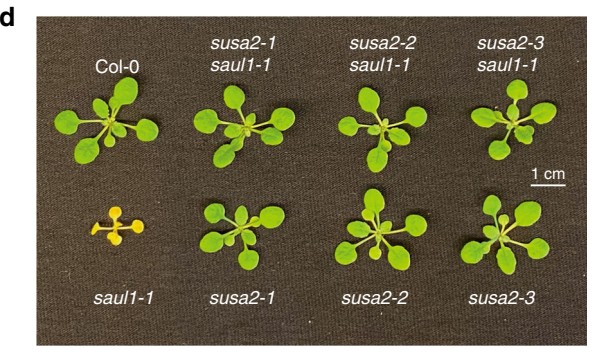

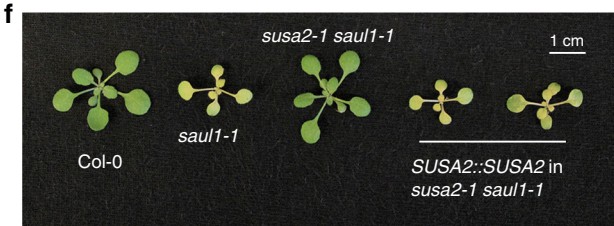

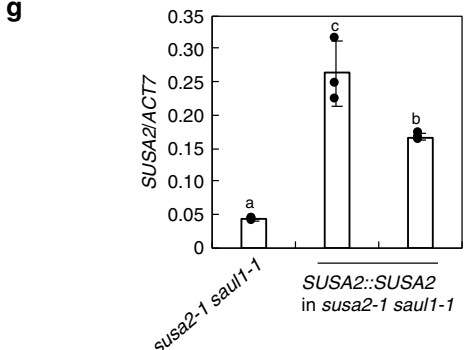

**Fig. 2 Positional cloning of *susa2-1*. a** Schematic diagram of chromosome 5 indicating the region where *susa2-1* was mapped to. The numbers of recombinants obtained during mapping are indicated in parentheses. **b** List of potential *susa2* candidate non-synonymous mutations in the mapped region obtained from next-generation whole-genome sequencing analysis. **c** The gene structure of *SUSA2*. Boxes and lines represent exons and introns, respectively. Gray boxes represent untranslated regions, and black ones represent coding sequences. The asterisk indicates the G-to-A mutation in *susa2-1*, and the arrows indicate the sites of T-DNA insertions in *susa2-2* and *susa2-3*. **d** Morphology of 3.5-week-old Col-0, *saul1-1*, *susa2-1 saul1-1*, *susa2-2 saul1-1*, *susa2-3 saul1-1*, and *susa2-1*, *susa2-2*, *susa2-3* plants. Plants were grown on 1/2 MS medium for 10 days and then transplanted on soil for two weeks before the picture was taken. **e** The predicted protein domains of SUSA2. The N-terminal F-box domain and C-terminal ACTIN domain are indicated with black boxes. The asterisk indicates the position of the conserved G252E amino acid (a.a.) change caused by the *susa2-1* mutation. **f** Morphology of 3.5-week-old Col-0, *saul1-1*, *susa2-1 saul1-1*, and two independent transgenic plants with *SUSA2::SUSA2* transformed into *susa2-1 saul1-1*. Plants were grown on 1/2 MS medium for 10 days and then transplanted on soil for two weeks before the picture was taken. **g** *SUSA2* expression in the indicated plants in **f** as determined by RT-PCR and normalized to *ACTIN7* (*ACT7*). Error bars represent means ± SD (one-way ANOVA, SPSS Statistics, *n* = 3, *p* < 0.01). Experiments were repeated three times with similar results.

phenotype of *saul1-1*. Transgene complementation was further performed to confirm the correct cloning of *SUSA3* by transforming the native *HSP90.3::HSP90.3* into *susa3-1 saul1-1* mutants[37]. As shown in Fig. 3d, e, *HSP90.3::HSP90.3* reverted *susa3-1 sau11-1* back to *saul1-1* phenotype. Taken together, *SUSA3* is *HSP90.3*.

**SUSA2 interacts with ASK1**. SCF E3 ubiquitin ligase is a multiprotein complex containing an F-box protein, S Phase Kinase-Associated Protein 1 (SKP1; ASK proteins in plants), Cullin1 (CUL1), and RING Box Protein 1 (RBX1)[34,38]. The F-box domain directly associates with SKP1/ASKs and the rest of the protein recruits substrate proteins to the SCF E3 ligase complex to be ubiquitinated. Therefore, substrate specificity of SCF is largely determined by the F-box protein[34]. The majority of ubiquitinated target proteins are subjected to 26S proteasome-mediated degradation. As SUSA2 contains a predicted F-box domain (Fig. 2e), we first tested whether SUSA2 could interact with ASK1 (Arabidopsis SKP1 Homolog 1) through a split-luciferase assay. As shown in Supplementary Fig. 4A–C, when *ASK1-CLuc* and *SUSA2-NLuc* were expressed together in *N. benthamiana* leaves, strong luminescence was observed. This interaction was further confirmed by co-IP assay in Arabidopsis transgenic lines carrying both *SUSA2-FLAG* and *ASK1-HA* (Supplementary Fig. 4D). Together, these data indicate that SUSA2 associated with ASK1 directly, and SUSA2 may indeed form part of an SCF E3 ligase complex.

BLAST analysis using SUSA2 protein sequence as input on The Arabidopsis Information Resource (TAIR) website revealed that *SUSA2* is a single-copy gene encoding a protein with an N-terminal F-box motif and a C-terminal ACTIN domain (Fig. 2e). In Arabidopsis, there are eight putative ACTIN-related proteins (ARPs), ARP2-ARP9, which are highly divergent among themselves and also divergent from conventional ACTINs, as revealed by protein sequence alignment (Supplementary Fig. 5)[31]. Although ARPs are found in all eukaryotes, phylogenetic analysis of SUSA2 and its paralogs showed that SUSA2, with the unique combination of F-box and ACTIN domains, is only found in the

revealing different roles of the chaperone in NLR activation and stability control. Serine 100 is located in the ATP-binding pocket of HSP90.3, a highly conserved region among HSP90s from yeast, human, *Caenorhabditis elegans* (*C. elegans*), and several plant species (Fig. 3b). To confirm that this mutation in *susa3-1* is responsible for suppressing *saul1-1*, we crossed the previously reported *hsp90.3-1* mutant with *saul1-1*[37]. As shown in Fig. 3c, as with *susa3-1*, *hsp90.3-1* largely suppresses the autoimmune

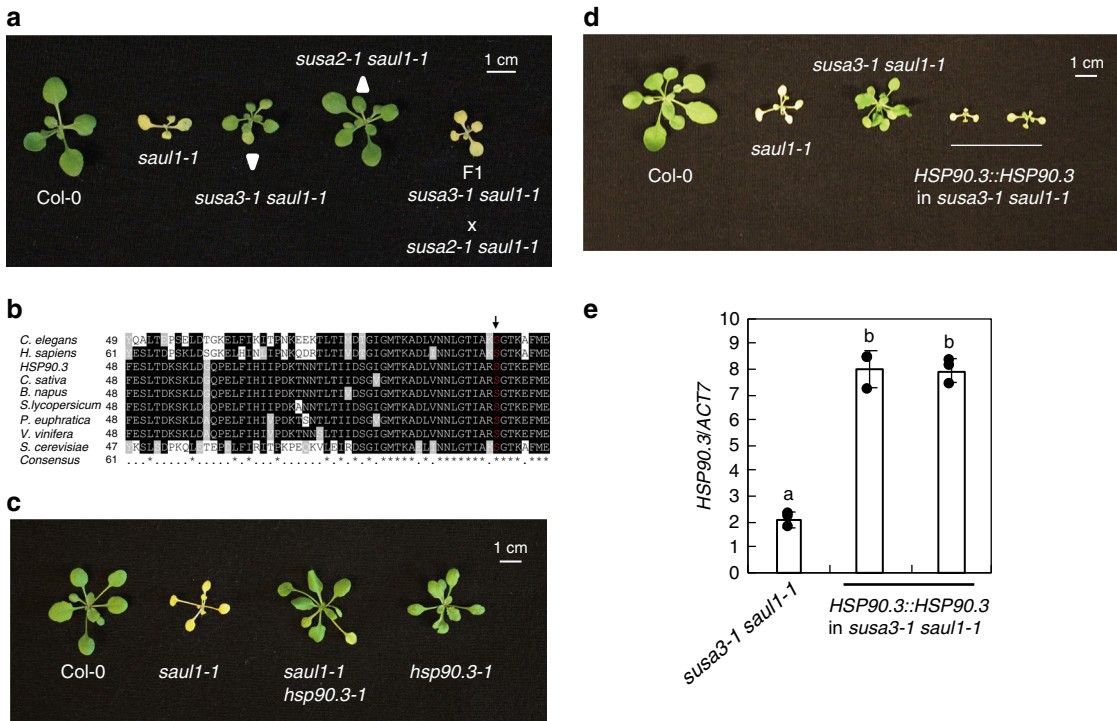

**Fig. 3 Characterization and positional cloning of *susa3-1*. a** Morphology of 3.5-week-old Col-0, *saul1-1* and *susa3-1 saul1-1*, *susa2-1 saul1*-1, and an F1 plant of *susa3-1 saul1-1* crossed with *susa2-1 saul1-1*. Plants were grown on 1/2 MS medium for 10 days and then transplanted to soil for 2 weeks before the picture was taken. **b** Part of the HSP90.3 protein alignment from multiple species. * indicates conserved amino acid. The arrow points to the mutated residue in *susa3-1*. **c** Morphology of 3.5-week-old Col-0, *saul1-1* and *saul1-1 hsp90.3-1*, and *hsp90.3-1* plants. Plants were grown on 1/2 MS medium for 10 days and then transplanted to soil for 2 weeks before the picture was taken. **d** Morphology of 3.5-week-old Col-0, *saul1-1*, *susa3-1 saul1-1*, and two independent transgenic plants with HSP90.3::HSP90.3 transformed into *susa3-1 saul1-1*. Plants were grown on 1/2 MS medium for 10 days and then transplanted to soil for 2 weeks before the picture was taken. **e** HSP90.3 expression in the indicated plants in **d** as determined by RT-PCR and normalized to ACT7. Error bars represent means ± SD (one-way ANOVA, SPSS Statistics, n = 3, p < 0.01). Experiments were repeated three times with similar results.

plant lineage (Supplementary Fig. 6)[39]. Protein alignment revealed that the ACTIN domain, which accounts for 70% of the SUSA2 amino acids, is highly conserved in SUSA2 paralogs throughout all examined plant species, but displays lower sequence identity with the ACTIN domains from human, yeast, *C. elegans,* and *Drosophila melanogaster* (Supplementary Figs. 7 and 8). The mutation site G252 in the ACTIN domain found in *susa2-1* is highly conserved in SUSA2 paralogs, suggesting its important function for SUSA2. As this domain likely interacts with the SCF substrate, this residue could be critical for binding with the unknown E3 target.

The ACTIN domain in SUSA2 shares only around 30% identity with conventional ACTINs. However, SUSA2 appears to maintain a general ACTIN fold for binding nucleotide, a common tertiary structure for conventional ACTINs (Supplementary Fig. 9A, B). ARPs' functions are hypothesized to be distinct from conventional ACTINs due to their divergent surface features. They have been implicated in ACTIN assembly and vesicle movement in the cytoplasm, and transcriptional regulation in the nucleus[39,40]. It is worth noting that this ACTIN fold is quite different from a typical nucleotide-binding (NB) domain of NLRs, such as the predicted one for SOC3 (Supplementary Fig. 9C).

To study the functional importance of the ACTIN domain of SUSA2, site-directed mutagenesis was firstly carried out to substitute conserved amino acids that are predicted to bind nucleotide (Supplementary Fig. 5). G258, K313, G399, G400 were mutated to alanines (Supplementary Fig. 10A). Both FLAG-tagged wild-type and mutant versions of SUSA2, including

SUSA2-FLAG and SUSA2-G258A-FLAG, SUSA2-K313A-FLAG, SUSA2-G399A/G400A-FLAG were transformed into *susa2-1 saul1-1*. As expected, SUSA2-FLAG fully complemented *susa2-1 saul1-1* (Supplementary Fig. 2A). To our surprise, all three mutant versions of SUSA2 still rescued *susa2-1 saul1-1* back to *saul1*-like morphology and displayed full complementation (Supplementary Fig. 10B, C). This indicates that G258, K313, G399, and G400 residues in the SUSA2 ACTIN domain are not required for SUSA2's function in *saul1-1*-mediated autoimmunity.

**susa2 mutants exhibit mild enhanced disease susceptibility toward a virulent *Pseudomonas* pathogen.** To further study the function of SUSA2, *susa2-1*, *susa2-2*, and *susa2-3* single mutants were generated and characterized. All mutant lines are indistinguishable from WT (Fig. 4a). When SUSA2 expression was measured in these lines, *susa2-2* and *susa2-3* showed a drastically reduced SUSA2 expression while *susa2-1* showed similar SUSA2 expression compared to WT (Fig. 4b), which is consistent with their loss-of-function lesions (Fig. 2). Upon infection by the virulent bacterial strain *P. syringae* pv. *maculicola* (*P.s.m.*) ES4326, *susa2* alleles exhibit slightly enhanced disease susceptibility (Fig. 4c), suggesting a minor role of SUSA2 in basal defense. When challenged with the avirulent pathogen *P. syringae* pv. *tomato* (*P.s.t.*) DC3000 expressing AvrRps4 and AvrRpt2, no significant difference in resistance was observed (Fig. 4d, e).

We also examined the role of SUSA2 in PTI by challenging *susa2* alleles with Type III Secretion System (T3SS) deficient

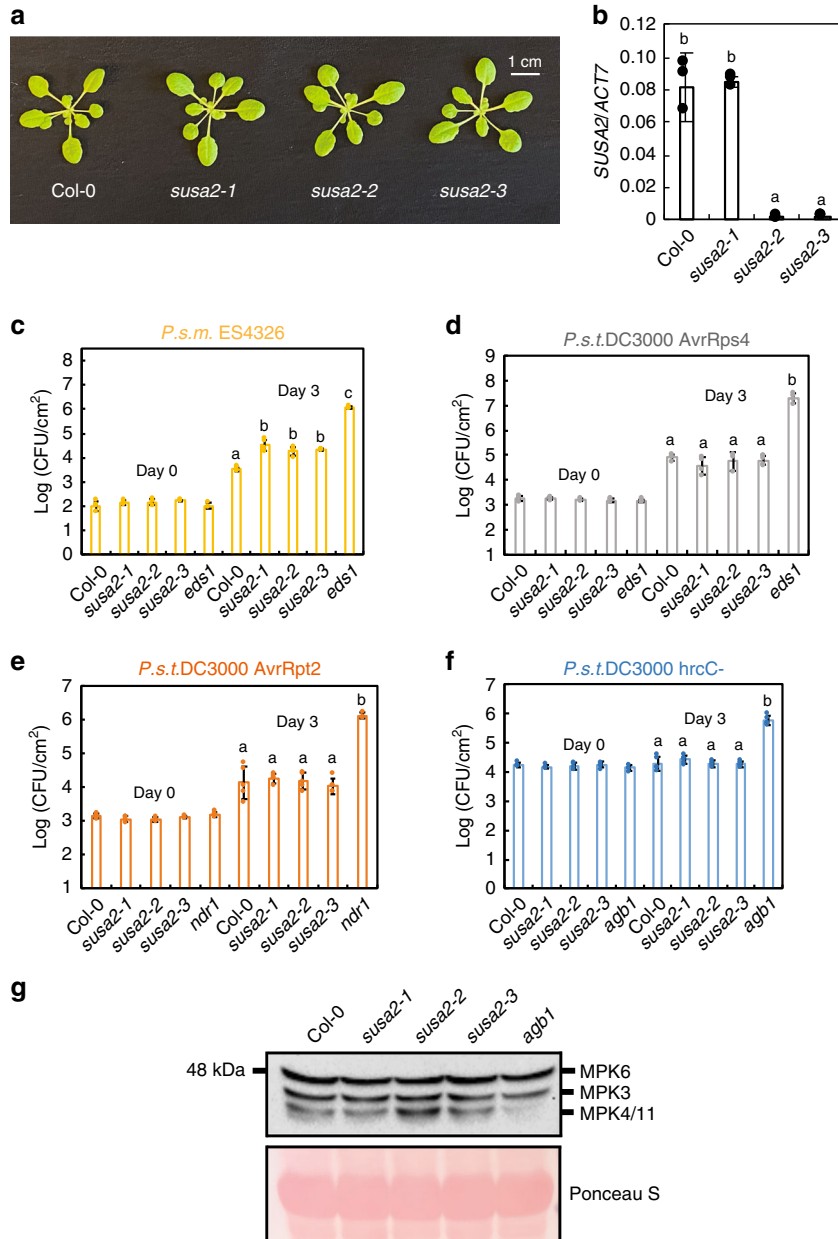

**Fig. 4 Characterization of the *susa2* single mutants. a** Morphology of 3.5-week-old Col-0, *susa2-1*, *susa2-2*, *susa2-3,* which were grown on soil. **b** *SUSA2* expression in the indicated plants as determined by RT-PCR and normalized to *ACT7*. Error bars represent means ± SD (one-way ANOVA, SPSS Statistics, $n = 3$, $p < 0.01$). Experiments were repeated three times with similar results. **c** Bacterial growth of *P.s.m.* ES4326 in Col-0, *susa2-1*, *susa2-2*, *susa2-3*, and *eds1* plants 3 days post infiltration. Error bars represent means ± SD (one-way ANOVA, SPSS Statistics, $p < 0.01$). For day 0, $n = 4$ for Col-0, *susa2-1*, *susa2-2*, and $n = 3$ for *susa2-3*, *eds1*. For day 3, $n = 5$ for Col-0, *susa2-2* and $n = 4$ for *susa2-1*, *susa2-3*, *eds1*. Experiments were repeated three times with similar results. *eds1* mutant shows high susceptibility to *P.s.m.*ES4326 and serves as a positive control. **d** Bacterial growth of *P.s.t.* DC3000 AvrRPS4 in Col-0, *susa2-1*, *susa2-2*, *susa2-3,* and *eds1* plants 3 days post infiltration. Error bars represent means ± SD (one-way ANOVA, SPSS Statistics, $n = 3$, $p < 0.01$). For day 0, $n = 4$. For day 3, $n = 4$ for Col-0, *susa2-1*, *eds1* and $n = 3$ for *susa2-2*, $n = 5$ for *susa2-3*. Experiments were repeated three times with similar results. *eds1* mutant shows high susceptibility to *P.s.t.* DC3000 AvrRPS4 and serves as a positive control. **e** Bacterial growth of *P.s.t.* DC3000 AvrRpt2 in Col-0, *susa2-1*, *susa2-2*, *susa2-3* and *ndr1* plants 3 days post infiltration. Error bars represent means ± SD (one-way ANOVA, SPSS Statistics, $n = 3$, $p < 0.01$). For day 0, $n = 4$. For day 3, $n = 5$ for Col-0, *susa2-1*, *susa2-2*, *susa2-3*, and $n = 3$ for *eds1*. Experiments were repeated three times with similar results. *ndr1* mutant shows high susceptibility to *P.s.t.* DC3000 AvrRpt2 and serves as a positive control. **f** Bacterial growth of *P.s.t.* DC3000 hrcC⁻ in Col-0, *susa2-1*, *susa2-2*, *susa2-3,* and *agb1* plants 3 days post infiltration. Error bars represent means ± SD (one-way ANOVA, SPSS Statistics, $n = 3$, $p < 0.01$). For day 0, $n = 4$. For day 3, $n = 5$ for Col-0, *susa2-1*, *susa2-3*, *eds1* and $n = 4$ for *susa2-2*. Experiments were repeated three times with similar results. *agb1* mutant shows high susceptibility to *P.s.t.* DC3000 hrcC⁻ and serves as a positive control. **g** flg22-induced MPK activation. 12-day-old plate-grown seedlings were sprayed with 1 μM flg22 for 15 min and then analyzed by immunoblots using anti-Erk antibody that recognizes the activated MAP kinases MPK6, MPK3, and MPK4/11. The experiment was repeated twice with similar results.

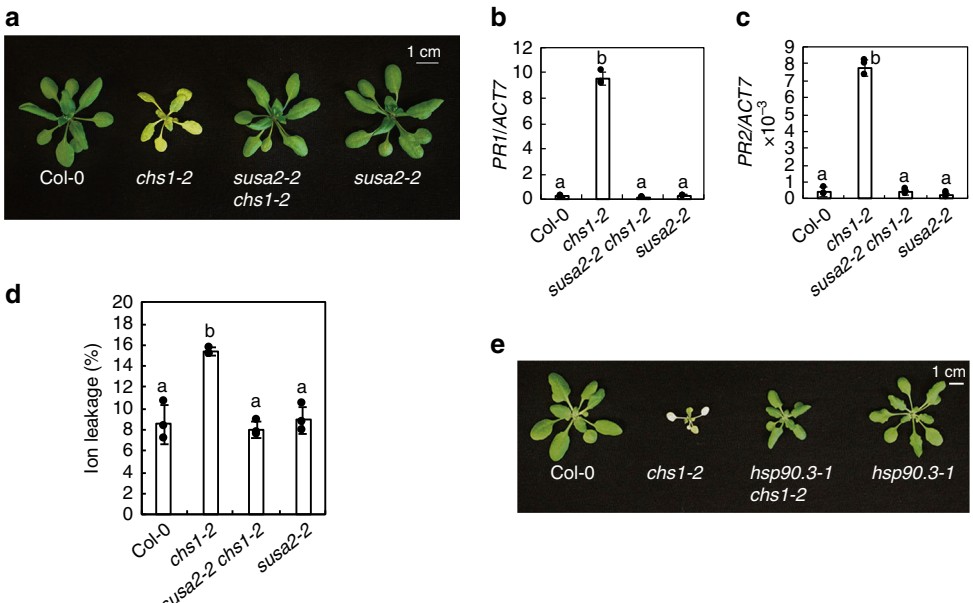

**Fig. 5 Both *susa2-2* and *hsp90.3-1* suppress the temperature-dependent autoimmunity of *chs1-2*. a** Morphology of 4-week-old Col-0, *chs1-2*, *susa2-2 chs1-2*, and *susa2-2*. Plants were grown on soil for 3 weeks and then moved to 18 °C for 1 week to induce the autoimmunity of *chs1-2*[26]. **b, c** RT-PCR analysis of the expression of *PR1* (**b**) and *PR2* (**c**) in the indicated genotypes. *ACT7* was used to normalize the transcript levels. Error bars represent means ± SD (one-way ANOVA, SPSS Statistics, $n = 3$, $p < 0.01$). Experiments were repeated three times with similar results. **d** Ion leakage measurement of the plants as shown in **a**. Ion leakage was calculated as the percentage of the conductivity before autoclaving over that after autoclaving. Eight leaf discs were used per genotype in 22.5 ml $H_2O$. Error bars represent means ± SD (one-way ANOVA, SPSS Statistics, $n = 3$, $p < 0.01$). Experiments were repeated three times with similar results. **e** Morphology of 4-week-old Col-0, *chs1-2*, *hsp90.3-1 chs1-2*, and *hsp90.3-1* plants. Plants were grown on soil for three weeks and then moved to 18 °C for 1 week before the picture was taken.

bacterial strain *P.s.t.* DC3000 hrcC⁻. As shown in Fig. 4f, no alteration of *P.s.t.* DC3000 hrcC⁻ growth was observed in the *susa2* alleles compared with WT. *susa2* lines also showed similar flg22-induced MPK activation as WT (Fig. 4g), confirming that SUSA2 does not contribute to PTI. Taken together, *susa2* mutants exhibit mildly enhanced susceptibility and SUSA2 has a positive role in plant immunity towards virulent pathogen *P.s.m.* ES4326.

**SUSA2 is specifically involved in SOC3-mediated immunity**. TNL protein SOC3 was previously reported to be required for the temperature-dependent autoimmunity of *chs1-2*[29]. To test whether SUSA2 contributes to *chs1-2*-mediated autoimmunity, *susa2-2 chs1-2* double mutant was generated. As shown in Fig. 5a, *susa2-2* largely suppresses *chs1-2*. RT-PCR analysis revealed that elevated expression levels of *PR* genes in *chs1-2* were greatly reduced by *susa2-2* (Fig. 5b, c). Ion leakage analysis confirmed that *susa2-2* reverted the cell death phenotype of *chs1-2* back to WT (Fig. 5d). As the HSP90 protein family is implicated in NLR-mediated immunity and *hsp90.3-1* can suppress *saul1-1*[15,16,37], *hsp90.3-1 chs1-2* double mutant was generated to test whether a mutation in *HSP90.3* may have any effect on *chs1-2*. As shown in Fig. 5e, *hsp90.3-1* partially suppresses the morphology of *chs1-2*. In summary, both SUSA2 and HSP90.3/SUSA3 are involved in SOC3–CHS1-mediated immunity.

*SAUL1* overexpression activates autoimmune responses, which are mediated by sensor NLR pair SOC3-TN2[24]. To examine whether SUSA2 contributes to SOC3-TN2-mediated immunity, the previously reported functional *35S::GFP-SAUL1* construct was transformed into Col-0 plants and *susa2-2* mutant background. Consistent with the previous findings, 8 out of 23 T1 transformants of *SAUL1* overexpression in Col-0 exhibit auto-immunity, including curly leaves and necrosis. However, in *susa2-2* mutant background, all 18 T1 transformants of *SAUL1*

overexpression are WT like and none display autoimmunity (Supplementary Fig. 11A, B), indicative of the requirement of SUSA2 for *SAUL1* overexpression autoimmune responses. Similarly, we also examined whether HSP90.3 is required for *SAUL1* overexpression mediated autoimmunity. When *35S::SAUL1 hsp90.3-1* double mutant was generated, its size is intermediate compared with both parents, suggesting a partial suppression of *35S::SAUL1* by *hsp90.3-1* (Supplementary Fig. 11C−D). Enhanced disease resistance against virulent oomycete pathogen *H.a.* Noco2 in *35S::SAUL1* was partially suppressed by *hsp90.3-1* (Supplementary Fig. 11E). Therefore, both SUSA2 and HSP90.3 also seem to be involved in SOC3-TN2-mediated immunity.

To further test the specificity of SUSA2 in plant immunity, we crossed *susa2-2* with a collection of autoimmune mutants including *snc1, snc2-1D, mekk1-5,* and *chs3-2D*. Defense responses are constitutively activated in *snc1*, which carries a gain-of-function mutation in a typical TNL[41,42]. *suppressor of npr1-1, constitutive 2* (*snc2-1D*), which encodes a Receptor-Like Protein (RLP) with a gain-of-function mutation in the transmembrane domain, is a unique genetic background to study signaling pathway downstream of RLPs[43]. In *mekk1-5* mutant, the CNL, Suppressor of *mkk1 mkk2*, 2 (SUMM2), is activated[44]. On the other hand, *chs3-2D* carries a mutation in an atypical TNL *CHS3*, which activates autoimmunity that is dependent on its typical TNL neighbor *CONSTITUTIVE SHADE-AVOIDANCE 1* (*CSA1*)[45,46]. As shown in Supplementary Fig. 12, *susa2-2* does not suppress the morphological phenotypes of *snc1, snc2-1D, mekk1-5*, or *chs3-2D*. Therefore, the autoimmunity-suppressing effect of *susa2-2* seems to be specific to *chs1-2, saul1-1*, and *SAUL1* overexpression. SUSA2 seems to be dedicated to TNL SOC3-mediated immunity.

**SUSA2 acts upstream of EDS1**. It was previously reported that *saul1-1* activates EDS1/PAD4-dependent defense responses[30]. To

examine the relationship between SUSA2 and EDS1, the *EDS1-YFP*[NLS] gain-of-function transgenic line, which displays severe autoimmunity due to expressing EDS1 with an SV40 nuclear localization signal (NLS)[47], was crossed with *susa2-2* to generate the double mutant. As shown in Supplementary Fig. 13, *susa2-2 EDS1-YFP*[NLS] resembles the parent *EDS1-YFP*[NLS] plant, indicating that *susa2-2* cannot suppress EDS1-conditioned autoimmunity and SUSA2 likely acts upstream of EDS1. Together, these genetic data place SUSA2 at the sensor NLR level during signaling.

**SUSA2 and SUSA3 associates with NLR pairs SOC3–CHS1 or SOC3–TN2.** Both animal and plant NLRs are known to oligomerize through the NB domain, and the P-loop within are often critical for NLR activation[20,48]. When the P-loop of CHS1 is mutated, it failed to complement *chs1-2* back to WT (Supplementary Fig. 14), confirming that the intact P-loop is essential for SOC3–CHS1 activation in *saul1-1*.

As NLRs are known chaperone clients, we speculated that SGT1b, together with HSP90.3, likely serve as chaperones for the assembly of an NLR activation complex containing CHS1 and SOC3. This is firstly supported by a previous report on SGT1b being involved in the SAUL1-mediated defense pathway, as an *sgt1b* mutant was identified as a genetic suppressor of *saul1-1*[27]. Such a hypothesis is also supported by our genetic data that *hsp90.3-1* suppresses the autoimmunity of *chs1-2* (Fig. 5e). To further test this hypothesis, physical interaction was examined by co-IP using transiently expressed HSP90.3 and SOC3 in *N. benthamiana*. As shown in Fig. 6a, HA-FLAG-SOC3 could pull down HA-HSP90.3, supporting that HSP90.3 functions as a molecular chaperone to assist with CHS1-SOC3 protein complex assembly.

In yeast, humans, and plants, SGT1 has also been shown to be a stable subunit of SCF E3 ligase complexes through its highly conserved interaction with SKP1/ASKs[49,50]. HSP90s are similarly known to be present in SCF E3 complexes to assist with complex assembly[51–53]. Therefore, the interactions between SGT1b/HSP90 and NLRs raise the possibility that NLR-mediated immunity may require physical association with an SCF E3 ligase complex. Since the F-box protein SUSA2, SGT1b, and HSP90.3 are all required for SOC3–CHS1-mediated autoimmunity in *saul1-1* mutant, we hypothesized that components of an SCF E3 ligase complex may associate with NLR pairs SOC3–CHS1 or SOC3–TN2.

To test this, we first tested whether SGT1b interacts with SUSA2. In a split-luciferase assay, SGT1b indeed interacted with SUSA2, suggesting that SGT1b may be part of the SCF[SUSA2] E3 ligase complex (Fig. 6b, c and Supplementary Fig. 15).

To further test the association between SUSA2 and NLRs, we examined whether CHS1 physically associates with SUSA2 in planta. As shown in Fig. 6d, the co-IP assay revealed that FLAG-tagged SUSA2 could pull down CHS1-HA, indicating that SUSA2 is also in complex with SOC3–CHS1. To independently test this interaction, we used a newly developed TurboID-based labeling method[54], where protein–protein interactions can be revealed by proximity-based biotinylation, and the biotinylated proteins due to protein–protein interactions can be detected by Streptavidin-HRP antibodies using western blot analysis. Such an unbiased method can reliably identify weak and transient interactions. As shown in Fig. 6e, immunoprecipitated DN-SUSA2-FLAG proteins could pull down CHS1-Turbo-HA, and DN-SUSA2-FLAG could be biotinylated by CHS1-Turbo-HA, further supporting the hypothesis that SUSA2 may interact with a SOC3–CHS1 NLR complex.

SUSA2–CHS1 interaction was further confirmed in co-IP assays using Arabidopsis stable transgenic lines carrying *SUSA2-FLAG*

and *CHS1-HA* (Fig. 6f). Interestingly, in *hsp90.3-1* or *sgt1b* mutant backgrounds, lower amounts of CHS1-HA were pulled down (Fig. 6f), in consistency with a chaperone function of HSP90 and SGT1 in facilitating large NLR protein complex assembly.

Since SUSA2 also contributes to SOC3-TN2-mediated autoimmunity in *SAUL1* overexpression lines, the association between SUSA2 and TN2 was tested by co-IP assay (Fig. 6g). DN-SUSA2-FLAG could be pulled down by TN2-FLAGTEVZZ, suggesting that SUSA2 can also interact with TN2. SUSA2-TN2 interaction was further confirmed in the co-IP assay using stable Arabidopsis transgenic lines carrying *SUSA2-FLAG* and *TN2-HA* and this interaction is likewise dependent on HSP90.3 and SGT1b (Fig. 6h). Similarly, the interaction between SOC3 and SUSA2 was tested using TurboID. As shown in Fig. 6i, HA-FLAG-SOC3 could pull down SUSA2, and SOC3 could also be biotinylated by DN-SUSA2-Turbo-HA. We also observed self-biotinylation of SUSA2, which is not unexpected, as F-box proteins in SCF complexes often exhibit self-oligomerization[55].

Lastly, since SUSA2 and ASK1 associate directly, we further tested our hypothesis that components of an SCF E3 ligase complex may associate with NLR pairs by examining whether ASK1 associates with SOC3. As shown in Fig. 6j, ASK1-FLAGTEVZZ pulled down HA-FLAG-SOC3 in a co-IP assay, further supporting the interaction between NLRs and components of an SCF E3 ligase complex. Taken together, all these pairwise protein–protein interaction data suggest that immunity mediated by plant NLRs requires components of the SCF[SUSA2] complex through physical protein–protein association (Fig. 7).

## Discussion

Microbial pathogens can be detected by NLRs in both mammals and higher plants. Despite their structural and functional similarities, mammalian NLRs are activated by conserved microbial PAMPs/MAMPs, rather than variable effectors as in higher plants[2]. This recognition difference partly explains the hugely expanded number and diversity of NLRs encoded in higher plant genomes. Such NLR variability and randomness of the effectors allude to likely differential rather than unifying, as in the case of mammalian NLRs, activation mechanisms under different recognition scenarios. The model of mammalian NLR activation is exemplified as NLR family apoptosis inhibitory protein 2 (NAIP2), which upon recognition of the bacterial rod protein PrgJ, forms a PrgJ-NAIP2-NLRC4 ($1 \times 1 \times 10$ ring) inflammasome to recruit and activate caspase-1 executor, turning on downstream immune responses[56]. For plant NLRs, recent structural data on ZAR1 resistosome reveals a similar pattern where perception of effector triggers the pentamerization of ZAR1 together with its adaptor protein RKS1 and decoy PBL2[10,11]. Other studies also support the general idea that plant NLRs need to oligomerize to function, in many cases with different NLRs in the same complex as in the case of RPS4-RRS1, RGA4-RGA5, SOC3–CHS1, and SOC3-TN2 pairs[19,21,22,24]. However, what happens upon NLR activation remains unclear. As there are no caspase-1 orthologs in higher plants, the executor equivalent of mammalian caspase-1 for plant immunity is missing in NLR resistosomes.

Here, our study suggests a model where an assembled SCF E3 ligase complex containing NLRs could be one way of NLR activation. In either *saul1-1* knockout mutant or *SAUL1* overexpression transgenic plants, NLR immune receptor pair SOC3–CHS1 or SOC3–TN2, respectively, is assembled and activated, resulting in autoimmunity[24,25]. The identification and detailed study of two additional suppressors of *saul1-1* uncovered that SUSA2, an F-box protein, which we propose contributes to SOC3-mediated immunity by forming the SCF[SUSA2] complex

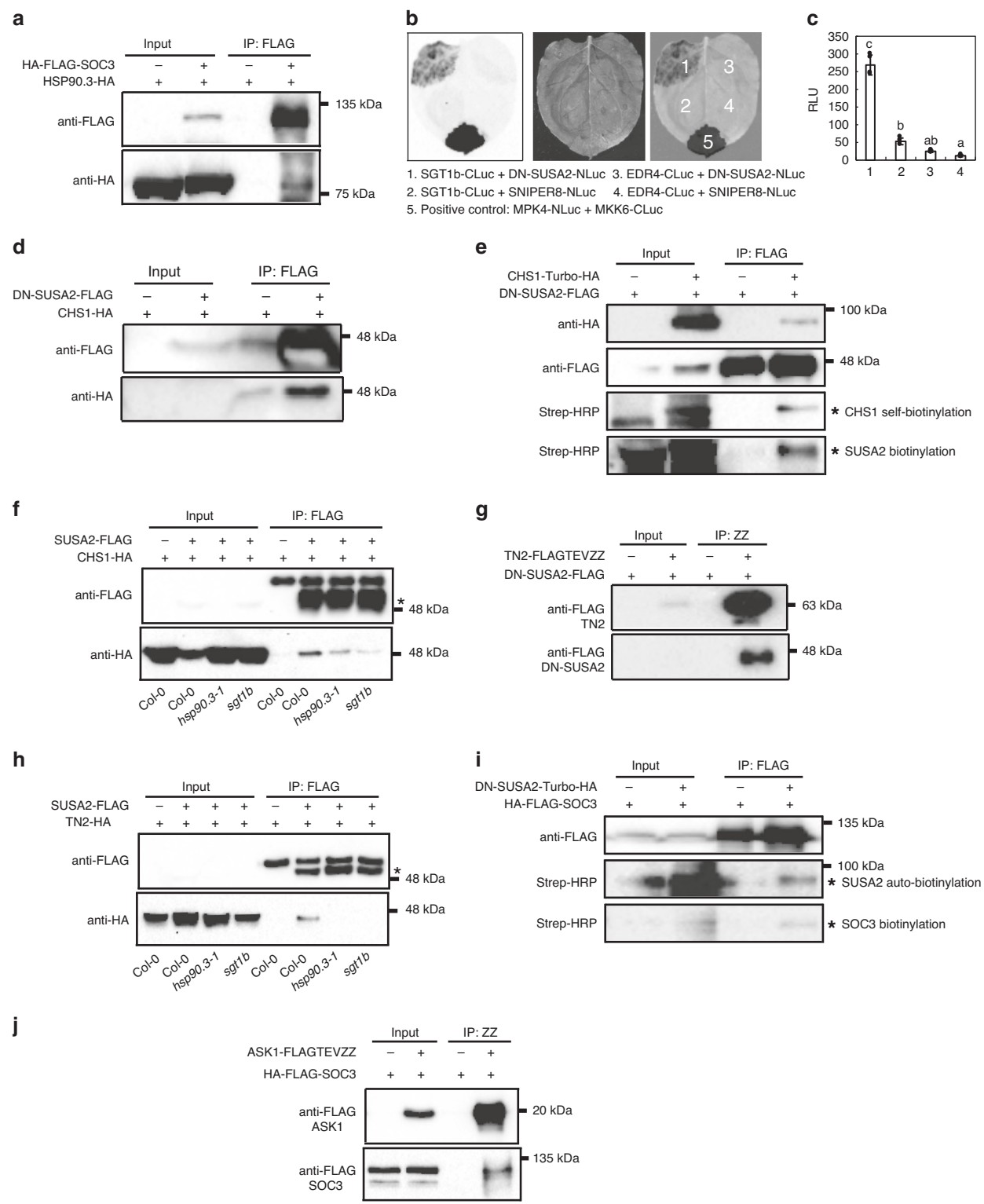

together with the NLR pairs (Fig. 7). All tested interactions in Fig. 6 occur before HR response, suggesting that these components might already exist in a preformed complex before NLR activation. The formation of such an NLR–SCF complex is likely facilitated and maintained by HSP90.3/SUSA3 and SGT1b chaperones. Upon proper assembly of an NLR–SCF complex, we suggest the E3 ligase may target an unknown E3 substrate, presumably either a negative regulator of defense for ubiquitination and degradation or a positive immune regulator for monoubiquitination and activation or subcellular localization

change, turning on immunity. Here, the E3 ligase would serve as an executor, with an analogous role to caspase-1 in the animal inflammasome model. However, at present, we have only tested pairwise interactions. Further work would be needed to test whether these proteins interact with each other simultaneously and form a large complex and to support the hypothesized executor role of the E3 ligase.

Recently, TIR domains of plant NLRs were demonstrated to possess NADase activity upon self-association, the products of which promote cell death during immune responses[57,58]. Since

**Fig. 6 Protein–protein interactions between the NLR pair SOC3–CHS1 or SOC3-TN2 and SCFSUSA2. a** HA-FLAG-SOC3 is able to pull down HSP90.3-HA. *Agrobacterium* carrying *HA-FLAG-SOC3* and *HSP90.3-HA* constructs or empty vector (−) were co-infiltrated into *N. benthamiana* leaves. After 48 h incubation, leaf tissue was harvested for co-immunoprecipitation analysis. **b** Split-Luciferase assay showing an interaction between SUSA2 and SGT1b. *Agrobacterium* carrying *SGT1b-CLuc* and *DN-SUSA2-NLuc* constructs or other indicated paired constructs were co-infiltrated into *N. benthamiana* leaves at $OD_{600} = 0.2$. Photos were taken 48 h post infiltration. MPK4-NLuc and MKK6-CLuc serve as positive control. ENHANCED DISEASE RESISTANCE4 (EDR4) is involved in negative regulation of resistance to powdery mildew[69]. SNIPER8 is an immune-regulating E3 ligase isolated from *snc1*-influencing plant E3 ligase reverse (SNIPER) genetic screen, the data of which has not been published (Paul Kapos and Xin Li). **c** Quantification of chemiluminescence in **b**. The luminescence of positive control was too high (8552 ± 474) and it masked the relatively weak interaction intensity between SGT1b and SUSA2. It was therefore not included in the graph. Error bars represent means ± SD (one-way ANOVA, SPSS Statistics, $n = 3$, $p < 0.01$). **d** DN-SUSA2-FLAG is able to pull down CHS1-HA. *Agrobacterium* carrying *DN-SUSA2-FLAG* and *CHS1-HA* constructs or empty vector (−) were co-infiltrated into *N. benthamiana* leaves. After 48 h incubation, leaf tissue was harvested for co-immunoprecipitation analysis. **e** CHS1-Turbo-HA can biotinylate DN-SUSA2-FLAG and be pulled down by DN-SUSA2-FLAG. *Agrobacterium* carrying *DN-SUSA2-FLAG* and *CHS1-Turbo-HA* constructs or empty vector (−) were co-infiltrated into *N. benthamiana* leaves. After 48 h incubation, leaf tissue was harvested for co-immunoprecipitation and subsequent western blot analysis. Asterisks indicate biotinylated protein bands. **f** SUSA2-FLAG is able to pull down CHS1-HA in *Arabidopsis* stable transgenic lines carrying the two transgenes. co-IP assays were conducted in Col-0, *hsp90.3-1*, and *sgt1b* backgrounds. Asterisks indicate SUSA2-FLAG protein bands. **g** DN-SUSA2-FLAG can be pulled down by TN2-FLAGTEVZZ. *Agrobacterium* carrying *DN-SUSA2-FLAG* and *TN2-FLAGTEVZZ* constructs or empty vector (−) were co-infiltrated into *N. benthamiana* leaves. After 48 h incubation, leaf tissue was harvested for co-immunoprecipitation analysis. **h** SUSA2-FLAG is able to pull down TN2-HA in *Arabidopsis* stable transgenic lines with both transgenes. co-IP assay was conducted in Col-0, *hsp90.3-1,* and *sgt1b* backgrounds. Asterisks indicate SUSA2-FLAG protein bands. **i** DN-SUSA2-Turbo-HA can biotinylate SOC3-HA-FLAG. *Agrobacterium* carrying *SOC3-HA-FLAG* and *DN-SUSA2-Turbo-HA* constructs or empty vector (−) were co-infiltrated into *N. benthamiana* leaves. After 48 h incubation, leaf tissue was harvested for co-immunoprecipitation and subsequent western blot analysis. Asterisks indicate biotinylated protein bands. **j** HA-FLAG-SOC3 can be pulled down by ASK1-FLAGTEVZZ. *Agrobacterium* carrying *HA-FLAG-SOC3* and *ASK1-FLAGTEVZZ* constructs or empty vector (−) were co-infiltrated into *N. benthamiana* leaves. After 48 h incubation, leaf tissue was harvested for co-immunoprecipitation analysis.

the TNLs CHS1, TN2, and SOC3 here all have TIR domains at their N-termini, we speculate that the TIR dimerization or oligomerization upon SOC3-CHS1 or SOC3-TN2 activation in *saul1-1* or *35S::SAUL1* plants, respectively, can trigger their NADase activity[24]. One possibility could be that one of the NADase products (not all have been defined clearly yet) may act as a glue to assist with this hypothetical NLR–SCF complex formation, similar to the binding of auxin to the SCF$^{TIR1}$ receptor complex to promote auxin response (Fig. 7a, b)[59]. An alternative hypothesis is that the induction of cell death by TIR NADase products is through an unknown molecular mechanism that is independent of the E3 ligase activity of the SCF$^{SUSA2}$ complex (Fig. 7c, d).

How generally can such model work in plant NLR activation? From the phenotypic analysis of *susa2* single mutants (Fig. 4), *susa2* plants exhibit mild susceptibility, disputing its general role in broad NLR activation. In addition, our genetic analysis indicates that SUSA2 seems to function specifically with the SOC3–CHS1 or SOC3-TN2 pairs (Figs. 1 and 5, Supplementary Figs. 11 and 12). Therefore, for this model to work generally, specific F-box proteins have to participate only in their cognate NLR activation. We believe such a model may be widespread for many plants NLRs due to the highly expanded F-box protein families encoded in the higher plant genomes, which coincides with the expansion of *NLR* genes. Plants may use different NLR–SCF complexes to target certain conserved key negative regulators of immunity for degradation, triggering a common defense output. Intriguingly, FBL41 was recently identified to function in race-specific immunity[60–62]. It could be possible that it may be involved in an NLR–SCF$^{FBL41}$ complex function with its unknown cognate NLR. Future discovery of more F-box proteins involved in other NLR activation would enhance such prediction.

Another support for such a model is the known general susceptibility of *hsp90* and *sgt1b* mutants, and the involvement of SGT1 and HSP90s in NLR activation complex assembly[13]. Intriguingly, both chaperones, especially SGT1, have been reported to be essential members of many different SCF complexes[50,52]. With our current model, SGT1 and HSP90 can be commonly involved in the assembly of many NLR–SCF complexes with different NLRs and distinct F-box proteins. Loss of these chaperones would then lead to disassembly of many NLR activation complexes, explaining their more general susceptibility phenotypes when their encoded genes are mutated.

One key missing player in this model is the unknown ubiquitination substrate of the SCF$^{SUSA2}$. In any SCF E3 ligase complex, the F-box protein is the major substrate determinant, with the F-box domain interacting with SKP1 and the rest of the protein binding to the substrate[34]. Therefore, the ACTIN domain of SUSA2 is predicted to be the direct ubiquitination target recognition domain. From our site-directed mutagenesis (Supplementary Fig. 10), G258, K313, G399, and G400 residues in the SUSA2 ACTIN domain are not required for SUSA2's function in *saul1-1*-mediated autoimmunity. Instead, its conserved ACTIN fold is likely required for substrate binding. From the phylogenetic analysis (Supplementary Figs. 6 and 7), *SUSA2* is a single-copy gene specifically found in plants. Therefore, it is possible that the substrate of SUSA2 is a conserved plant-specific negative regulator of immunity, its ubiquitination and degradation can lead to strong immunity. It is equally possible that the activated SCF$^{SUSA2}$ may monoubiquitinate its positive immune regulatory substrate, activating immunity. We tried to search for such substrate(s) using IP coupled to mass spectrometry (IP-MS) proteomics with DN-SUSA2 as the bait. However, due to the very low abundance of the bait protein, our IP-MS did not yield any significant candidates. Future attempts with modern approaches feasible for low abundance, transient, or weak protein–protein interactions such as TurboID-based proximity labeling may help us identify the missing substrate of SUSA2.

## Methods

**Plant materials and growth conditions**. Plant materials in this study include *Arabidopsis thaliana* and *Nicotiana benthamiana*. Unless specified, all plants were grown at 22 °C in growth rooms under a 16-h-light/8-h-dark regime. Plants for infection assays were grown at 22 °C under an 8-h-light/16-h-dark regime. As *saul1-1* plants are seedling lethal at 22 °C[25,30,63], for seeds collection, *saul1-1* and *saul1*-like plants were grown at 27 °C to suppress their autoimmunity.

**The *saul1-1* suppressor screen and NGS whole-genome sequencing**. The *saul1-1* suppressor screen was described previously[25]. For whole-genome sequencing, the genomic DNA of the *susa2-1 saul1-1* was extracted and purified using a Qiagen plant

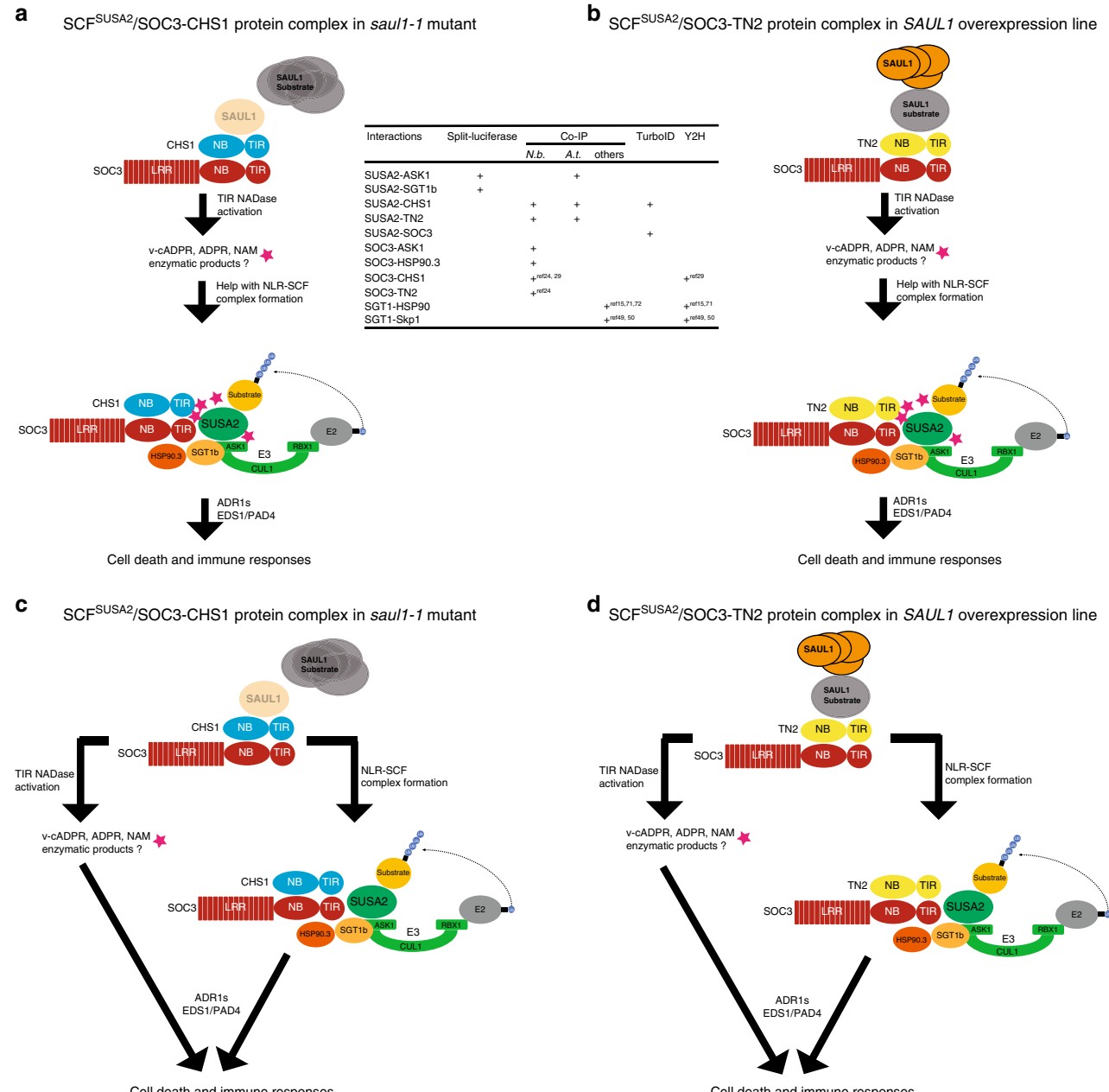

**a** SCF$^{SUSA2}$/SOC3-CHS1 protein complex in *saul1-1* mutant

**b** SCF$^{SUSA2}$/SOC3-TN2 protein complex in *SAUL1* overexpression line

**c** SCF$^{SUSA2}$/SOC3-CHS1 protein complex in *saul1-1* mutant

**d** SCF$^{SUSA2}$/SOC3-TN2 protein complex in *SAUL1* overexpression line

| Interactions | Split-luciferase | Co-IP | | | TurboID | Y2H |
|---|---|---|---|---|---|---|
| | | *N.b.* | *A.t.* | others | | |
| SUSA2-ASK1 | + | | + | | | |
| SUSA2-SGT1b | + | | | | | |
| SUSA2-CHS1 | | + | + | | + | |
| SUSA2-TN2 | | + | + | | | |
| SUSA2-SOC3 | | | | | + | |
| SOC3-ASK1 | + | | | | | |
| SOC3-HSP90.3 | + | | | | | |
| SOC3-CHS1 | +[ref24, 29] | | | | | +[ref29] |
| SOC3-TN2 | +[ref24] | | | | | |
| SGT1-HSP90 | | | | +[ref15,71,72] | | +[ref15,71] |
| SGT1-Skp1 | | | | +[ref49, 50] | | +[ref49, 50] |

**Fig. 7 The hypothetical assembly of the SOC3–CHS1 or SOC3-TN2 NLR protein complex with SCF$^{SUSA2}$.** This working model illustrates the assembly of the SCF$^{SUSA2}$ E3 ligase complex with the SOC3-CHS1 or SOC3-TN2 NLR pairs in *saul1-1* mutant (**a**, **c**) or *SAUL1* overexpression line (**b**, **d**), respectively. The middle table summarizes all the pairwise protein–protein interaction data supporting such a model. "+" represents positive protein–protein interactions. Blank space means not tested. The references are indicated as shown in the figure[15,24,29,49,50,71,72]. Here, NLR pairs SOC3–CHS1 and SOC3-TN2 guard the homeostasis of E3 ligase SAUL1, where SOC3–CHS1 is activated upon SAUL1 disappearance, while SOC3-TN2 is triggered upon *SAUL1* overexpression[24,25]. Activation of such NLR-SCF protein complex leads to the ubiquitination and degradation of the substrate of the SCF$^{SUSA2}$ E3 ligase, which is currently unknown. SOC3–CHS1 or SOC3-TN2 pairs may not necessarily exist as dimers; they more likely form oligomers as in the case of ZAR1 resistosome. During NLR activation, TIR domain dimerization or oligomerization of either SOC3-CHS1 or SOC3-TN2 in *saul1-1* or *35S::SAUL1* plants turn on TIR NADase activity, leading to accumulating of its products[57,58]. Under a linear model (**a**, **b**), one of the TIR products may act as a glue (star in the figure) to help with NLR–SCF complex formation, similar to the binding of auxin to the SCF$^{TIR1}$ receptor complex to promote auxin responses (**a**, **b**)[59]. However, a branching model is equally possible (**c**, **d**), where the induction of cell death by TIR NADase products is through an unknown molecular mechanism, which is independent of the E3 ligase activity in NLR–SCF complex. The assembly of the SCF$^{SUSA2}$–NLR complex triggers downstream defense activation, which is relying on EDS1, PAD4, and helper NLRs ADR1s[28,30,70].

DNA extraction kit. The library preparation and Illumina sequencing were performed by BGI (Beijing Genomic Institute, Beijing, China).

**Gene expression analysis**. Total RNA was extracted from 50 mg plant tissue, either 12-day-old 1/2 MS medium-grown seedlings or 4-week-old soil-grown plants using the EZ-10 Spin Column Plant RNA Mini-Preps Kit (Bio Basic, Canada). 2 μg RNA was reversely transcribed to cDNA using Easy Script$^{TM}$ reverse transcriptase (ABM, Canada). 50 ng cDNA was added as a template in a 10 μl reaction on a Bio-Rad CFX Connect$^{TM}$ Real-Time system machine. Real-time PCR was conducted to quantify the relative expression level of the target genes. *ACTIN1/7* was used to normalize the expression value.

**Construction of plasmids**. For the construction of *pCambia1305 SUSA2::SUSA2*, *SUSA2* genomic fragment with its native promoter, which includes 1.5 kb region upstream ATG start codon, was amplified from purified Col-0 WT genomic DNA using primers SUSA2-EcoRI_F and SUSA2-KpnI_R and ligated into *pCambia1305* vector after digestion with *Eco*RI/*Kpn*I. SUSA2-FLAG construct was generated using primers SUSA2-KpnI_F and SUSA2-BamHI _R. SUSA2-G258A-FLAG, SUSA2-K313A-FLAG, SUSA2-G399A/G400A-FLAG constructs were made by overlapping PCR using primers SUSA2_G258A_F, SUSA2_G258A_R, SUSA2_-K313A_F, SUSA2_K313A_R, SUSA2_G399A/G400A_F, SUSA2_G399A/G400A_R. Dominant-negative (DN)-SUSA2 genomic fragment, in which the F-box domain was deleted by overlapping PCR using primers SUSA2-F-box-dele_F and SUSA2-F-box-dele_R, was ligated into *pCambia1300-3FLAG* vector after digestion with *Kpn*I/*Bam*HI to generate DN-SUSA2-FLAG construct using primers SUSA2-KpnI_F and SUSA2-BamHI _R. Please refer to Supplementary Table 1 for primer details of other constructs.

**Ion leakage measurement**. Ion leakage measurement was performed according to a previously described protocol with some modifications[64]. Briefly, for each genotype, eight leaf discs were punched using rosette leaves from four plants and placed in a 50 ml conical tubes. There were three replicates for each genotype. 25 ml deionized water was added into each tube and shaken overnight. The conductivity of the solution was measured using a VWR Portable Conductivity Meter, Model 2052. The tubes with the leaf discs were then autoclaved. After cooling down to room temperature, the conductivity of the solution was measured again. Ion leakage was calculated as the percentage of the conductivity before autoclaving over that after autoclaving.

**Trypan blue staining**. Trypan Blue staining was performed according to a previously described protocol[65]. Briefly, the trypan blue staining solution was prepared by mixing solution (10 ml glycerol, 10 ml lactic acid, 10 g phenol, 10 mg trypan blue, and 10 ml water) with ethanol at a 1:1 ratio. The 14-day-old *Arabidopsis* seedlings were submerged in 1.0 ml trypan blue staining solution and then boiled for 2 min. After removing the staining solution, 2 ml chloral hydrate solution (1.25 g/ml) was added to destain the samples on a shaker overnight. Photos were taken by the camera using Dinocapture 2.0 software.

**Infection assays**. For oomycete pathogen *H.a.* Noco2 infection assays, seeds of the indicated genotypes were sterilized and planted on soil and grown in a growth room under a 16 h light:8 h dark cycle for 2 weeks before spraying with *H.a.* Noco2 conidiospore suspension. For *H.a.* Noco2 infection assay in which *saul1-1* plants were included, seeds were grown on half Murashige and Skoog (1/2 MS) medium for 7 days in a growth chamber at 22 °C and then transplanted on soil and grown in a growth chamber at an elevated temperature at 27 °C for another 4 days. Plants were then spray-inoculated with *H.a.* Noco2 conidiospore suspension at a concentration of 100,000 spores/ml, covered and grown with 80% humidity in a growth chamber at 18 °C. After 7 days, *H.a.* Noco2 growth was quantified by counting spores on leaf surface using a light microscopy per mg Fresh Weight (FW) following a protocol described in ref. [66].

For bacterial pathogen *P. syringae* infection assays, 4-week-old plants were used and leaves were syringe-infiltrated with a bacterial suspension at $OD_{600} = 0.001$. Bacterial growth was quantified by counting colony-forming-units (cfu) at 0 days post inoculation (dpi) and 3 dpi, respectively, as described previously[41].

**Split-luciferase assay**. The split-luciferase assay was performed based on a previously described protocol[67]. Briefly, the cloned CLuc and NLuc constructs were transformed into *Agrobacterium* and transiently expressed in *N. benthamiana* leaves by co-infiltration. After 48 h, the freshly prepared luciferin solution (1 mM Sigma D-luciferin prepared in DMSO, 10 mM MgCl₂, 10 mM MES/KOH buffer pH 5.6) was then infiltrated into *N. benthamiana* leaves, where CLuc- and NLuc-fused proteins were expressed. The fluorescence signal was detected and quantified on a Bio-Rad gel documentation system. MPK4-NLuc and MKK6-CLuc serve as positive control[68].

**Protein transient expression in N. benthamiana**. For transient expression in *N. benthamiana*, *Agrobacterium* carrying the indicated cloned constructs were cultured in LB liquid medium with appropriate antibiotics in a 28 °C shaker for 16 h and then cells were collected by centrifugation, transferred to the resuspension medium (4.5 g/l KH₂PO₄, 10.5 g/L K₂HPO₄, 1.0 g/l (NH₄)₂SO₄, 0.5 g/l sodium citrate, 0.5% glycerol, 0.2% glucose, 50 μM acetosyringone, 1 mM MgSO₄, and 10 mM N-morpholino-ethanesulfonic acid (MES) pH 5.6). After 8-hr growth in a 28 °C shaker, cells were collected by centrifugation and resuspended in the infiltration buffer (4.4 g/l MS powder, 10 mM MES, 150 μM acetosyringone) and infiltrated at a dosage of $OD_{600} = 0.2$.

**Total protein extraction and co-immunoprecipitation assay**. For plant total protein extraction, around 100 mg plant tissue was collected into 2.0 ml tubes with glass beads inside, frozen in liquid nitrogen, and ground into a fine powder using a grinding machine. 0.1 ml freshly made protein extraction buffer (0.1 M Tris-HCl pH 8.0, 0.1% SDS, 2% ß-mercaptoethanol) was then added to the samples, vortexed, and incubated on ice for 5 min. After centrifugation at $12,000 \times g$ for 5 min, the supernatant was transferred to a new set of 1.5 ml tubes with 4× SDS protein loading buffer added (60 mM Tris-HCl pH 6.8, 2% SDS, 10% glycerol, 5% ß-mercaptoethanol, and 0.01% bromophenol blue) and boiled for 5 min at 95 °C before subject to the following western blot analysis by using primary antibodies anti-HA (11867423001, Roche), anti-FLAG (Cat. #F1804, Sigma) at 1:2000 dilution and secondary antibodies goat anti-mouse (32230, Thermo Fisher), goat anti-rat (2065, Santa Cruz) at 1:5000 dilution. Biotinylated proteins in TurboID-based labeling method were checked by Streptavidin-HRP (Abcam Cat. # ab7403) at 1:20000 dilution.

For the co-immunoprecipitation assay, around 2 g *Agrobacterium*-infiltrated *N. benthamiana* leaf tissue was harvested 48 h after transient expression and ground into a fine powder with a set of pre-chilled mortar and pestle with liquid nitrogen. Around 5 ml protein GTEN buffer (10% glycerol, 25 mM Tris-HCl pH 7.5, 1 mM EDTA, 150 mM NaCl, 0.15% NP-40*, 10 mM DTT*, 2% PVPP, protease inhibitor cocktail from Sigma*, 1 mM PMSF*, * freshly added before use) was then added and incubated for 10 min with gentle shaking in a 4 °C cold room. The mixture was then centrifuged at 14,000 rpm for 5 min at least twice in order to get clear supernatant. FLAG-tagged proteins were immunoprecipitated by incubating the supernatant with 20 μl anti-FLAG M2 beads from Sigma for 3 h at 4 °C and then pelleted down by centrifugation for 1 min at $6000 \times g$ and washed five times with gentle shaking using washing buffer (10% glycerol, 25 mM Tris-HCl pH 7.5, 1 mM EDTA, 150 mM NaCl, 0.15% NP-40*, 10 mM DTT*, 1 mM PMSF*, * freshly added before use). 4× SDS protein loading buffer was added to the anti-FLAG beads and boiled for 5 min at 95 °C before subject to western blot analysis using the corresponding antibodies.

**MPK activation assay**. 12-day-old seedlings grown on 1/2 MS medium plates under long day conditions (16 h light/8 h dark) were sprayed with 1 μM flg22 plus 0.01% Silwet L-77 and incubated for 15 min before harvesting. Protein bands were analyzed by immunoblots using an anti-Erk primary antibody (Cell signaling; #4370 S) at 1:2000 dilution and goat anti-rabbit secondary antibody (2030, Santa Cruz) at 1:5000 dilution.

**Reporting summary**. Further information on research design is available in the Nature Research Reporting Summary linked to this article.

## Data availability
The authors declare that the data supporting the findings of this study are available within the article and its Supplementary Information files. Sequencing data associated with this study have been deposited to the NCBI Sequence Read Archive with accession number PRJNA664448. Source data are provided with this paper.

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

## Acknowledgements

We thank Kevin Ao (University of British Columbia) for critical reading of the manuscript. We thank Dr. Jane E. Parker (Max-Planck Institute for Plant Breeding Research) for sharing *EDS1-YFP^{NLS}* seeds. We thank Dr. Christoph Ringli (University of Zürich) for kindly sharing *der1* seeds. Dr. Wei Li (Hunan Agricultural University) is thanked for helping with Sanger sequencing and Dr. Tongjun Sun is thanked for help with quantification of chemiluminescence in split-luciferase assay. The Arabidopsis Biological Resource Center is acknowledged for providing all T-DNA mutant seeds. W.L. is partly supported by a scholarship from China Scholarship Council (CSC). Research described is funded to X.L. by grants from NSERC-Discovery and CREATE (PRoTECT) programs, and CFI-JELF.

## Author contributions

Under the supervision of X.L., W.L. performed most of the experiments and made the figures. M.T. performed the *saul1-1* suppressor screen and isolated *susa2-1* and *susa3-1* mutants. W.L. and X.L. wrote the manuscript. All authors reviewed the manuscript.

## Competing interests

The authors declare no competing interests.
