## [Peer Review File · Nature Communications]

Reviewers' comments:

Reviewer #1 (Remarks to the Author):

Review of manuscript 225220_0_merged_1567548033 submitted to Nature communications entitled:

Assembly of SCFSUSA2 E3 ligase is required for immunity mediated by 2 paired NLRs SOC3-CHS1 or SOC3-TN2

by Wanwan Liang, Meixuezi Tong and Xin Li

In the presented manuscript the authors identified and characterized two new suppressors of the Arabidopsis autoimmune mutant saul1-1. One of the suppressors, *susa3-1*, is a new allele of the well known immune-/NLR-regulator HSP90.3, a chaperone protein most likely involved in NLR homeostasis. The other suppressor mutant, *susa2-1*, encodes for a thus far uncharacterized plant-specific ACTIN-domain containing F-BOX protein named SUSA2. The authors provide data indicating that the TIR-NLR protein SOC3, of which the mutant also suppresses saul1-1 autoimmunity (Tomg et al., 2017 and Liang et al 2019), is part of a SCF-E3 ligase complex, consisting of HSP90.3(SUSA3)/SGT1b/SUSA2 and either one of the two truncated TN proteins TN2 or CHS1. The authors hypothesize that the assembly of this SCFSUSA2 E3 ligase complex is required for the activation of the 'bound' (guarding?) NLRs (see the title and lines 95-97) and functions as a 'resistosome' in degrading putative negative regulators of immunity (Figure 7). The figures are very well presented and the genetic data is convincing. However, necessary controls are missing in certain experiments (see below) and some experimental results are over interpreted and should be reconsidered or tuned down in the discussion (see also below). In addition, do the authors discuss important results that are not presented in the figures and therefore not part of the manuscript.

Nonetheless, the presented genetic data and the resulting hypothesis are of interest for the plant and animal NLR field and probably also everyone working in the field of plant immunity.

Unfortunately, I do not see a significant breakthrough for the NLR-field in the current form of the manuscript (in terms of convincing results to support the idea of an activation SCF E3 ligase NLR complex reminiscent of the animal NLR inflammasome activation). Also, because the *susa2-2* mutant seems to specifically suppress the SOC3 mediated auto-immune phenotypes and no other NLR or RPL (*snc2-1D*) mediated auto-immune phenotype (Figure S10). Their hypothesis of a more general function of F-BOX proteins (and therefore SCF E3 ligase complexes) in NLR activity (NLR-mediated immunity) is in my opinion not well supported by the presented results and as such only, a rather interesting, hypothesis.

I do not see this manuscript, in its current form, as being suitable for being published in Nature communications.

Major and minor issues:

Line 1 (title): I am not sure whether the title reflects the data. In my opinion the authors do not show that specifically the assembly of the SUSA2 containing SCF complex is really required and not rather the mere presence is really required for immunity. Additionally, no immunity against pathogens was tested to be affected by a loss of this complex (or SUSA2), thus the title should rather say auto-immunity mediated by paired NLRs SOC3-CHS1 or SOC3-TN2.

Lines 113-115 and Figure 1B: Do the saul1-1 plants have spontaneous lesion formation/cell death that can be visualized by trypan blue staining of the leaves? If so, is this also rescued/compromised in the *susa2* and *susa3* saul1 double mutants? Why is the Ion leakage results for WT so variable  compare value of WT in figure 3 with experiment shown in figure 5?

Figure 1a and Figure 2F: Authors show images of WT, saul1-1 mutant and saul1-1 suppressed plants as well as the *susa2-2* single mutant (Figure 2F) in these two figures. Comparing the *susa2-1* saul1-1 double mutant phenotype (Figure 1a) with the *susa2-2* saul1-1 double mutant (Figure 2F) they do not look too similar, the *susa2-2* saul1-1 double mutant much more resembles the

susa3-1 saul1-1 double mutant – is this just by chance or does the susa2-2 allele not rescue fully, as compared to the susa2-1 allele?

Additionally, comparing the susa3-1 saul1-1 and the susa2-2 saul1-1 phenotypes (using the provided plant images) with the 35S:SAUL1 over expressor phenotype as shown in (Liang et al., 2019 New Phytologist) they look pretty similar. Could there be a link between the susa3-1 (and the susa2-2) suppression of saul1-1 and this SAUL1 overexpressor phenotype?

Figure 2B: The table would benefit from indicating that At5g56180 is SUSAA2 and that the protein domain structure in 2E is SUSAA2.

Lines 151/152: Authors write that the SUSAA2 protein abundance is extremely low and thus they use a dominant-negative version of SUSAA2 for their interaction analysis. Authors should provide data supporting this claim.

Lines 157-159 and Figure S2: Authors present a Co-IP experiment to analyse interaction of SUSAA2 and SAUL1, by pulling immunoprecipitating DN-SUSAA2-FLAG. They do not observe/detect any interaction with HA-SAUL1. Was this IP also done in the other direction? Sometimes there is no interaction observable in one direction, but in the other - this lab should know this, since they experienced a similar result in their previous New Phytologist paper -Liang et al., 2019. It would be good to at least try the IP in the other direction.

Lines 201-203: This assay does not necessarily show direct interaction - it is still an in planta assay where a 'bridging' protein could be between ASK1 and SUSAA2.

Do the authors have another experiment to prove direct interaction, for example GST pulldowns of recombinant protein? Otherwise, I would phrase this claim differently and say just an interaction is indicated. Could one use a 'generic substrate' to show ubiquitination by this SCFSUSAA2 E3 ligase complex in vitro?

Lines 216-218: Authors write that the mutation of G252 in the ACTIN domain of SUSAA2 is highly conserved and thus explains the loss-of-function phenotype. I would not agree; the mere conservation of an amino acid does not explain a loss-of-function phenotype. It suggests an important function for the protein, but this is no proof and therefore no explanation. Also, it is interesting that here a mutation in the actin domain is found to compromise SUSAA2 function, but (according to the authors) other mutations of conserved residues in the ACTIN domain do not affect SUSAA2 function (figure S9B).

Lines 221-222 and figure S8: Maybe the authors could indicate where the nucleotide would bind in the structural model. Could this ACTIN domain also adopt a NB-ARC like tertiary structure?

Line 233-234 and Figure S9B: Here the authors claim that the SUSAA2 mutants with the mutations in the ACTIN domain fully complement the saul1-1 susa2-1 phenotype back to saul1-1 phenotype. However, to me this is not a full complementation at all and rather a 'new' phenotype. Do the authors also have generated susa2-1 plants and if so, what is the phenotype of the susa2-1 plants and the susa2-1 plants transgenic with the SUSAA2-FLAG wt and mutants? Since the expression of non-tagged SUSAA2 in the saul1-1 susa2-1 mutant fully complements the phenotype back to wt (figure 2c), I wonder whether there is a tag-effect caused by the FLAG tag and the expression of SUSAA2-FLAG causes a new phenotype? Also, it is not clear to me what promoter was used for the SUSAA2-FLAG constructs.

Lines 244-246: Please provide images showing the non-complementation of the ACTIN domain swapping construct in order to compare them to the other complementation. In addition, it would be great to see western blots showing expression of used constructs for any experiment presented (transgenic, complemented plant lines and transient expressed proteins in the split-luciferase assay.).

Lines 250-256: The authors write that two other *susa2* alleles (T-DNA alleles, *susa2-2* and *susa2-3*) were isolated and characterized. However, beside the presented pathogen infection phenotypes tested no image of the mutant plants in comparison to *susa2-1* are shown. Also, no proof of the knockout of the said lines is provided (RT-PCR showing that these alleles are indeed null alleles, especially since the two T-DNA lines used are depicted as intron insertions). Further it would be nice to show that all three *susa2* alleles suppress *saul1-1* autoimmunity similar.

Figure 4: Why was the originally identified *susa2-1* allele not used side-by-side with the two T-DNA alleles in this experiment?

Also, in Figure 4 the *eds1*, *ndr1*, *agb1* and *sid2* mutants were used as highly susceptible mutants (controls) – it should be mentioned what these mutants are – at least in the figure legend or the material and method part.

Lines 272-273 and figure 5b: In the figure the PR1 expression in the *susa2-2* allele is also higher than in the wt, was this a consistent phenotype? It would be nice if the authors could shortly discuss this also in the manuscript. Further, it would be good to provide statistics for the results in Figure 5b and c.

Figure 5A and E: I am not sure whether this is the normal variability of the *chs1-2* mutant phenotype, but the *chs2-1* mutants in figure 5A and 5F look quite different, also they were grown under the same conditions, right. Is this the normal phenotypical variation?

Figure 5D: authors present Ion leakage data obtained similar to what is shown in figure 1B, but the percentage presented here are super low compared to figure 1B. Can the authors comment on that please?

Lines 264-265: The authors write that, in light of their pathogen infection assays, *SUSA2* plays a positive role in plant immunity against virulent pathogens. I would agree in terms of the one tested (*Psm ES4326*), but this is the only one tested, thus I suggest to specifically mention this.

Lines 282-290: Authors describe a whole set of experiment(s) – generating transgenic plants overexpressing 35S::GFP-SAUL1 in Col-0 and the *susa2-2* mutant background and discuss the obtained data to make the point that *SUSA2* is also required for the SOC3-TN2 mediated 35S::SAUL1 autoimmune phenotype. However, no data is presented for this experiment(s). To make such a point, it would be great to see the actual data, including images of the transgenic plants and either western blots or RT-PCRs showing the expression of the transgene in the specific backgrounds and phenotypes (i.e. no or only a weak expression in non-phenotypic plants vs. high or detectable expression in phenotypic plants).

Lines 313-315: Authors present an experiment where they wanted to check whether the *susa2-2* mutant can suppress the EDS1-YFP-NLS auto-immune phenotype and come to the result that the *susa2-2* mutant cannot suppress this phenotype – indicating that *SUSA2* acts upstream of EDS1. Given the data obtained from the earlier mentioned experiments in the manuscript and from Disch et al., (2016) this was to be expected, since the dependency of SOC3 (and the autoimmune mutants that require SOC3 presence) are all EDS1 dependent.

Line 321: Authors write here that the "SOC3-CHS1 NLR complex is constitutively activated in *saul1-1* background..." - Is it really clear whether these NLRs are really 'activated'? The experiments done here and in the past (as far as I can remember) just demonstrate that their presence is required for the phenotype. Can a SOC3 or CHS1 P-loop mutant complement their knock-out, for example in the *saul1-1 soc3* or *saul1-1 chs1-2* mutants, respectively? I think in terms of what we know thus far about activity of NLRs required for (auto-)immunity, we should be more specific in this regard and not talk too specifically about their activation when this was not tested.

Lines 335-336: Is this interaction experiment really a proof showing that SGT1b for being part of an SCF SUSA2 E3 ligase complex? I think it just indicates it, but does not show it! The only thing it shows is that SUSA2 can interact with SGT1b in transient overexpression in *N. benthamiana* in a split-luciferase assay.

Figure 6C: Why is the positive control used in figure 6B not included in figure 6C?

Figure 6D (and E): authors show their Co-IP results where they 'show' interaction of DN-SUSA2-FLAG with CHS1-HA (6D) and TN2-FLAGTEVZZ with DN-SUSA2-FLAG. In figure 6D there are also bands in the negative control IP lane for both proteins or at least at a similar size. Could the authors please comment on that. How can they be sure that what they see here is really an interaction? Also, where is CHS1 in the input samples in figure 6D and where is DN-SUSA2 in the input samples in figure 6E? Maybe authors could provide an image of the whole membrane and an longer exposure to show their presence in the input.

Line 372: Authors write here that the assembly of the SCFSUSA2 E3 ligase complex facilitated by the NLRs SOC3 and either TN2 or CHS1 is resulting in constitutive ETI (effector-triggered immunity). Is this really an active process of assembling and activation, or is this complex anyways assembled and through (outside) disruption activated? Also, I would not write 'constitutive ETI', since there is no evidence of a pathogenic effector protein involved.

Line 375: The manuscript has no Figure 8.

Lines 379-380: here the authors write: "The assembly of such complex is likely facilitated and maintained by HSP90.3/SUSA3 and SGT1b chaperons." This is confusing, since in line 369 the authors write that the NLRs facilitate the assembly of this complex. Who is now responsible of the complex formation? Please try to make this clear.

Lines 378-380: The conclusions the authors make here are not supported by there data – at least in my view. Their hypothesis implies that this SCFSUSA2 E3 ligase complex is not formed in plants/cells that are not undergoing an immune response (or where SOC3/TN2/CHS1 NLRs are not activated), right? To confirm such a hypothesis the authors should perform Co-IP experiments with double transgenic Arabidopsis plants expressing functional tagged proteins for example in wildtype and saul1-1 background - here they should be able to see complex formation only in the saul1-1 mutant and not in wildtype. Otherwise the presented data does not, in my view, support such a conclusion/hypothesis or comparison with the animal NLR inflammasome recruiting caspsase-1.

Lines 403-405: The authors write: "..., the ACTIN domain of SUSA2 cannot function as ACT2 and unlikely serves a canonical ACTIN function." I do not agree that this interpretation of the experiments provided is correct, also part of the experiment results is not presented and rather only mentioned. The SWAP experiment only shows that replacing the ACTIN domain of SUSA2 with that of ACTIN2 results in a non-functional SUSA2 protein not having the ability to complement the susa2-2 mutant. A function of the SUSA2 ACTIN domain in complementing a actin2 mutant was not done.

Lines 406-408: Where is the data showing that SUSA2 gene 'appearance' coincides with the appearance of NLR genes in plant lineages? The 'evolutionary analysis in figure S4 and S5 do not provide this data.

Figure S2: Authors provide an image of a Co-IP between HA-SAUL1 and DN-SUSA2-FLAG and see no interaction of DN-SUSA2-FLAG with HA-SAUL1. Was this IP done in a reciprocal manner as well with similar results? Could the C-terminal FLAG tag have an influence on this interaction? NOTE that the complementation of the susa2-2 saul1-1 by the FLAG-tagged SUSA2 was not complete as well – at least in my point of view.

Figure S3: here is also the positive control value missing in panel B.

Figure S6: It would be great if the authors could also include the sites of the introduced point mutations that are supposed to affect canonical ACTIN function.

Figure S8: Could the authors indicate the residues important for nucleotide binding, as mentioned in the text and provide the PDB number of the structure used to model SUSAA2 actin domain.

Reviewer #2 (Remarks to the Author):

The plant immune system relies on intracellular NLR receptors to directly and indirectly detect pathogen virulence effectors. Exactly how these receptors function remains obscure. Xin Li's group has expanded the number and diversity of NLR mechanisms by screening for suppressors of autoactive NLRs. This has led to some interesting observations where full-length TIR-NBS-LRR (TNL) receptors require "truncated" partner TIR-NB (TN) genes. The exact mechanism by which hetero-NLR receptors function is unclear. In this work, the authors report on suppressor mutants that block *saul1-1* autoactive cell death. SAUL1 is an E3 ligase that appears to be guarded by a combination of the TNL SOC3 and genomically neighboring TN genes CHS1 and TN2. The hypothetical pathogen effector that triggers the SOC3/TN NLR system(s) remains unknown. The suppressors encode a previously isolated mutation in HSP90.3 and a new F-box protein SUSAA2. The repeated recovery of specific HSP90 alleles is an interesting observation, and it's still unclear how these alleles differentially affect various NLRs. The SUSAA2/ARP8 is an interesting suppressor with a fascinating domain structure. The mechanism is unclear and not really tested here, but hopefully there will be follow-up stories.

I have no major reservations about this paper and think it can be published in essentially its current form with edits to the text. I think that discussing HSP90 S100F in more detail (see comments below) would improve the paper by more explicitly tying it back to the literature. The text discussing the model figure and the figure itself could be more explicit about the supporting data and its limitations.

Specific comments:

Line 138) Need to introduce the F-box motif here to help the reader.

Line 138) Add references for ARP8

Check text for missing articles: e.g. Line 139) "A transgene complementation...", Line 141) "When the genomic region...", Line 143) "...displayed a *saul1*-like phenotype".

Line 153) List AA deleted in dnSUSAA2 construct.

Line 177) S100F in HSP90.3 same position as S100F allele of HSP90.2 previously found to destabilize RPM1 (Hubert 2003). HSP90.2 and HSP90.3 are 99% identical. You should comment more on the unusual recessive GOF phenotype of HSP90s in regards to destabilizing NLRs. If you have tested an HSP90.3 KO mutant, it would be expected to not suppress? Also worth testing for loss of RPM1 function in *hsp90.3* S100F (Ah, I now see this is all tested in your Huang et al MUSE paper as you independently isolated S100F again!?) You should include SUSAA3 in the discussion of this paper to comment on this more explicitly, currently there is nothing?

Line 181) Worth mentioning that previously isolated *hsp90.3-1* is the same mutation (S100F) as *susa3-1*.

Line 237) "Therefore the ACTIN domain does not serve a canonical ACTIN function" This seems over-interpreted. We don't know that the hypothetical ACTIN function is required to suppress saul1-1. It's possible that SUSAA2 protein accumulation is sufficient for saul1-1 suppression rather than SUSAA2 function. What SUSAA2 does in other contexts is unknown and not assayed. Seems safer to just say that these residues are not required for SUSAA2 function to suppress saul1-1.

Line 327-345; Figure7) Seems like the model is extremely hypothetical. Can be presented more clearly by denoting pairwise interactions/evidence? Data presented are mostly pairwise coIP, but this is not strong evidence for a multiprotein complex (beyond pairwise interactions)? So if A coIPs B and B coIPs C, that is not sufficient information to say if there is an A/B/C complex, or if there are independent A/B and B/C complexes.

Discussion: More discussion of a possible non-degradation/proteasome function for ubiquitination by SUSAA2/ARP8? ARP8 previously localized to nucleolus (Kandasamy, 2008), is nucleolar function consistent with what is known about SOC/CHS1/TN2 localizations?

Figure 7: Don't call this a resistosome. Resistosome implies a ZAR1-like oligomer, no data presented to support this?

Figure 7: Flip position of SOC3 and TIR-NBs as no coIP evidence for TIR-NBs and HSP/SGT? E3 oval too large, just outline E3 components?

Reviewer #3 (Remarks to the Author):

The authors previously showed that loss of function of the PUB44 E3 ligase (SAUL1) leads to activation of TIR-NLR pair SOC3-CHS1, whereas overexpression of PUB44 leads to activation of TIR-NLR pair SOC3-TN2, suggesting that SOC3 pairs with CHS1 or TN2 to monitor PUB44 homeostasis. The authors hypothesize that the aforementioned TIR-NLR pairs monitor a substrate (substrates) of PUB44. This study aims to identify this substrate by genetic screen. The authors successfully identified two genes whose mutations suppressed autoimmune phenotype in the pub44 background in which SOC3 is known to be activated. One gene encodes HSP90.3, the other, called SUSAA2, encodes an F-box protein with a C terminal extension sharing modest homology with actin. Further genetic analyses confirmed that SUSAA2 is required for both SOC3-CHS1 and SOC3-TN2 function, but not other NLRs, indicating that SUSAA2 is specifically required for SOC3 function. Genetic analysis also suggested that SUSAA2 acts upstream of EDS1. The authors further show that SUSAA2 can interact with TN2, CHS1, ASK1, and SGT1, and HSP90.3 can interact with SOC3. The authors conclude that the aforementioned TIR-NLR pairs and SUSAA2 are assembled into a large SCF complex and that this process is facilitated by HSP90.3 and SGT1. The authors also suggest that this complex allows the TIR-NLR pairs to execute signaling by ubiquitinating and degrading negative regulators of defenses. While the genetic analyses are sound and solid, the protein analyses are insufficient to support the conclusions. Also, the proposed model is at odds with current understanding of TIR-NLRs.

Major comments:

1. Based on pair-wise protein-protein interaction experiments, the authors suggest that SOC3, CHS1, HSP90, SGT1b, SUSAA2 and ASK1 all exist in the same protein complex. It is at least equally possible that they exist in different protein complexes. Can TN2 or CHS1 pull-down ASK1 or CUL? Can ASK1 pull-down TN2, CHS1, or SOC3? Can these proteins be detected in the same complex via gel filtration or blue native gel? Comprehensive analyses are needed before such conclusion can be made.
2. Although PUB44 (SAUL1) interacts with the two TIR-NLR pairs, it does not interact with SUSAA2. This further argues against the idea that these proteins exist in the same complex. A technical

issue: A truncated SUSAA2 protein lacking the F-box was used for protein-protein interaction studies. Although this increases protein level, it is difficult to draw conclusion from a lack of interaction between this construct and PUB44. If the F-box happens to be the interaction domain, then the construct will not be able to detect the interaction.

3. The authors suggest that HSP90.3 and SGT1b facilitate the assembly of NLR-SCF complex.

What about the protein levels of SOC3, CHS1, and TN2? In addition to SCF complex assembly in yeast and animals, HSP90 and SGT1b are known to be involved in stability control of NLRs in both plants and animals. Also, it is necessary to show the SOC3-CHS1, SOC3-TN2, CHS1/TN2-SUSAA2, and SUSAA2-ASK1 interactions are affected by mutations in HSP90.3 and SGT1b. If HSP90.3 and SGT1b are required for the assembly of the NLR-SCF protein complex, mutations of HSP90.3 and SGT1b are expected to impair the interactions.

4. The study started with PUB44, but their final model does not have a place for PUB44. Do PUB44 and SUSAA2 affect each other in protein stability? Could PUB44 be a substrate for SUSAA2?

5. Additional evidence is needed to support the idea that an intact SCF complex is required for the function of two TIR-NLR pairs.

6. Even if a SCF complex is involved, it is problematic to envision the SCF complex as an executor of immune signaling. Recent studies show that the TIR domain acts as a NADase which presumably release an unknown product as a signal to trigger the EDS1-dependent immunity, which is executed by NRG1 and ADR1 proteins. The idea that SUSAA2-mediated ubiquitination and degradation of a negative regulator of defense does not explain the previous understanding of TIR-NLR signaling. Also, SUSAA2 is only required for SOC3 but not other NLR function. Most likely interpretation would be that SUSAA2 is monitored by SOC3-CHS1/TN2 pairs.

7. In all split-luc experiments, empty vectors were used as negative controls. nLUC vector does not contain a start codon, and it provides a false negative. This must be corrected.

Minor comments:

1. Lines 216-218: "The mutation site G252 in the ACTIN domain found in susa2-1 mutant is also highly conserved, explaining its loss-of-function when mutated". Any suggestion why this site affects function? Other conserved residues involved in nucleotide binding are not required.

2. Does SUSAA2 play a role in PTI? The authors measured bacterial growth for hrcC mutant. It is necessary to thoroughly measure PTI responses.

3. In discussion, the comparison with caspase in inflammasome may not be appropriate. Again, there is no evidence that SUSAA2 is an executor. The fact that it is specifically required for SOC3 but not other NLR function suggests that it is likely a guardee, not an executor. What is the phenotype of SUSAA2 overexpression plants?

4. Line 375, "Fig 8" should be "Fig 7".

5. Fig S5, the analysis should include lower plants in which TIR-NLRs first appear.

We sincerely thank the time and efforts of the three expert reviewers in reviewing our
manuscript, which helps us to improve the quality, thoroughness and readability of our story. We
carefully addressed almost all the criticisms raised, which lead to substantial additions and
revisions. Here our revision details can be found in blue font underneath each original comment.

We apologize for the delay in resubmitting the revision, as we were not allowed to do
experiments for a few months due to COVID-19 pandemic. Even nowadays, we are only allowed
partial occupancy, and researchers have limited time to spend in the lab. The graduation timeline
for the first author PhD student Wanwan Liang has been delayed as a consequence, since the
revised manuscript forms a major chapter of her thesis.

Reviewers' comments:

Reviewer #1 (Remarks to the Author):

Review of manuscript 225220_0_merged_1567548033 submitted to Nature communications
entitled:

Assembly of SCFSUSA2 E3 ligase is required for immunity mediated by 2 paired NLRs SOC3-
CHS1 or SOC3-TN2

by Wanwan Liang, Meixuezi Tong and Xin Li

In the presented manuscript the authors identified and characterized two new suppressors of the
Arabidopsis autoimmune mutant saul1-1. One of the suppressors, susa3-1, is a new allele of the
well-known immune-/NLR-regulator HSP90.3, a chaperone protein most likely involved in NLR
homeostasis. The other suppressor mutant, susa2-1, encodes for a thus far uncharacterized plant-
specific ACTIN-domain containing F-BOX protein named SUSA2. The authors provide data
indicating that the TIR-NLR protein SOC3, of which the mutant also suppresses saul1-1
autoimmunity (Tomg et al., 2017 and Liang et al 2019), is part of a SCF-E3 ligase complex,
consisting of HSP90.3(SUSA3)/SGT1b/SUSA2 and either one of the two truncated TN proteins
TN2 or CHS1. The authors hypothesize that the assembly of this SCFSUSA2 E3 ligase complex
is required for the activation of the 'bound' (guarding?) NLRs (see the title and lines 95-97) and
functions as a 'resistosome' in degrading putative negative regulators

of immunity (Figure 7). The figures are very well presented and the genetic data is convincing.
However, necessary controls are missing in certain experiments (see below) and some
experimental results are over interpreted and should be reconsidered or tuned down in the
discussion (see also below). In addition, do the authors discuss important results that are not
presented in the figures and therefore not part of the manuscript.

Nonetheless, the presented genetic data and the resulting hypothesis are of interest for the plant
and animal NLR field and probably also everyone working in the field of plant immunity.

Unfortunately, I do not see a significant breakthrough for the NLR-field in the current form of
the manuscript (in terms of convincing results to support the idea of an activation SCF E3 ligase
NLR complex resistosome reminiscent of the animal NLR inflammasome activation). Also,
because the susa2-2 mutant seems to specifically suppress the SOC3 mediated auto-immune
phenotypes and no other NLR or RPL (snc2-1D) mediated auto-immune phenotype (Figure S10).
There hypothesis of a more general function of F-BOX proteins (and therefore SCF E3 ligase
complexes) in NLR activity (NLR-mediated immunity) is in my opinion not well supported by

the presented results and as such only, a rather interesting, hypothesis.
I do not see this manuscript, in its current form, as being suitable for being published in Nature
communications.

Major and minor issues:

Line 1 (title): I am not sure whether the title reflects the data. In my opinion the authors do not
show that specifically the assembly of the SUS A2 containing SCF complex is really required and
not rather the mere presence is really required for immunity. Additionally, no immunity against
pathogens was tested to be affected by a loss of this complex (or SUS A2), thus the title should
rather say auto-immunity mediated by paired NLRs SO3-CHS1 or SOC3-TN2.

The title has been changed accordingly.

New title: SCF^{SUS A2} E3 ligase is required for autoimmunity mediated by paired NLRs SOC3-
CHS1 or SOC3-TN2

Lines 113-115 and Figure 1B: Do the *saul1-1* plants have spontaneous lesion formation/cell
death that can be visualized by trypan blue staining of the leaves? If so, is this also
rescued/compromised in the *susa2* and *susa3* *saul1* double mutants? Why is the Ion leakage
results for WT so variable  compare value of WT in figure 1 with experiment shown in figure
5?

We carried out trypan blue staining to address this. As shown in Figure 1B, the staining showed
that cell death in *saul1-1* is partially and completely suppressed in *saul1-1 susa3-1* and *saul1-1*
*susa2-1* mutants, respectively.

For the mentioned ion leakage assays, different numbers of punched leaf discs in 25ml deionized
water were used, leading to difference in the absolute conductivity values. Detailed explanation
is now added to the figure legend to avoid confusion.

Figure 1a and Figure 2F: Authors show images of WT, *saul1-1* mutant and *saul1-1* suppressed
plants as well as the *susa2-2* single mutant (Figure 2F) in these two figures. Comparing the
*susa2-1 saul1-1* double mutant phenotype (Figure 1a) with the *susa2-2 saul1-1* double mutant
(Figure 2F) they do not look too similar, the *susa2-2 saul1-1* double mutant much more
resembles the *susa3-1 saul1-1* double mutant – is this just by chance or does the *susa2-2* allele
not rescue fully, as compared to the *susa2-1* allele?

Additionally, comparing the *susa3-1 saul1-1* and the *susa2-2 saul1-1* phenotypes (using the
provided plant images) with the 35S:SAUL1 over expressor phenotype as shown in (Liang et al.,
2019 New Phytologist) they look pretty similar. Could there be a link between the *susa3-1* (and
the *susa2-2*) suppression of *saul1-1* and this SAUL1 overexpressor phenotype?

The plants in pictures of the previous Fig 1A and 2F were grown at different times, showing
slightly different morphology due to mild growth condition differences. We re-grew the plants
and took a new picture of *susa2-2 saul1-1* plant in Figure 2D, which were grown under more
similar conditions as Fig 1A. As all three alleles of *susa2* can fully suppress *saul1*, they are likely
all null alleles.

*susa3-1* shows a partial suppression of *saul1-1* (Figure 1). We do not believe there is any link
between the *susa3-1* suppression of *saul1-1* and the SAUL1 overexpression phenotype the
reviewer refers to. The observed curly leaf phenotype of these plants is quite common for
autoimmune Arabidopsis mutants. For example, *snc1* can look quite similar to SAUL1
overexpression lines under certain growth conditions. Although the size of the plants always
remains small for autoimmune mutants, their leaf morphology can vary slightly each time.
Therefore, the plant morphology is not a good indication for judging the autoimmunity sources.

Figure 2B: The table would benefit from indicating that At5g56180 is SUS A2 and that the
protein domain structure in 2E is SUS A2.

For clarity, we added the indication as suggested.

Lines 151/152: Authors write that the SUS A2 protein abundance is extremely low and thus they
use a dominant-negative version of SUS A2 for their interaction analysis. Authors should provide
data supporting this claim.

As shown in the new Figure S2C, Agrobacteria carrying *SUS A2-FLAG* or *DN-SUS A2-FLAG*
constructs were infiltrated into left or right half of one leaf, respectively, with only *DN-SUS A2-*
*FLAG* expressed at a reasonable level in three replicates. The *SUS A2-FLAG* protein cannot be
detected using western blot, although it is able to complement the *susa2* mutation (Figure S2A).

Lines 157-159 and Figure S2: Authors present a Co-IP experiment to analyse interaction of
SUS A2 and SAUL1, by pulling immunoprecipitating *DN-SUS A2-FLAG*. They do not
observe/detect any interaction with *HA-SAUL1*. Was this IP also done in the other direction?
Sometimes there is no interaction observable in one direction, but in the other - this lab should
know this, since they experienced a similar result in their previous New Phytologist paper -Liang
et al., 2019. It would be good to at least try the IP in the other direction.

As suggested by the reviewer, we carried out the reciprocal IP, the data of which can be found in
Figure S3B. Consistently, *SAUL1-C29A-FLAG* could not pull down *SUS A2-HA*.

Lines 201-203: This assay does not necessarily show direct interaction - it is still an in planta
assay where a 'bridging' protein could be between *ASK1* and *SUS A2*.
Do the authors have another experiment to prove direct interaction, for example GST pulldowns
of recombinant protein? Otherwise, i would phrase this claim different and say just an interaction
is indicated. Could one use a 'generic substrate' to show ubiquitination by this SCFSUS A2 E3
ligase complex in vitro?

We carried out split-luciferase assay to test direct protein-protein interaction between *ASK1* and
*SUS A2* (Figure S4), which confirms the *ASK1-SUS A2* interaction.

Unlike simple E3s such as the RING type, where in vitro ubiquitination can be routinely
conducted, SCF E3s are complex E3s requiring many subunits, chaperons, and adaptors. There

are currently technical difficulties to reliably carry out *in vitro* ubiquitination assay for plant SCF
E3s.

Lines 216-218: Authors write that the mutation of G252 in the ACTIN domain of SUSAA2 is
highly conserved and thus explains the loss-of-function phenotype. I would not agree; the mere
conservation of an amino acid does not explain a loss-of-function phenotype. It suggests an
important function for the protein, but this is no proof and therefore no explanation. Also, it is
interesting that here a mutation in the actin domain is found to compromise SUSAA2 function, but
(according to the authors) other mutations of conserved residues in the ACTIN domain do not
affect SUSAA2 function (figure S9B).

As suggested, the statement has been changed to “The mutation site G252 in the ACTIN domain
found in *susa2-1* mutant is also highly conserved, suggesting its important function for SUSAA2”.

Lines 221-222 and figure S8: Maybe the authors could indicate where the nucleotide would bind
in the structural model. Could this ACTIN domain also adopt a NB-ARC like tertiary structure?

The nucleotide binding area is now indicated in Figure S9A. Structural comparison between
SUSAA2 ACTIN domain with the predicted NB-ARC domain of NLR protein SOC3 was
conducted, revealing low structural similarity as shown in Figure S9C.

Line 233-234 and Figure S9B: Here the authors claim that the SUSAA2 mutants with the
mutations in the ACTIN domain fully complement the *saul1-1 susa2-1* phenotype back to *saul1-1*
phenotype. However, to me this is not a full complementation at all and rather a 'new'
phenotype. Do the authors also have generated *susa2-1* plants and if so, what is the phenotype of
the *susa2-1* plants and the *susa2-1* plants transgenic with the SUSAA2-FLAG wt and mutants?
Since the expression of non-tagged SUSAA2 in the *saul1-1 susa2-1* mutant fully complements
the phenotype back to wt (figure 2c), I wonder whether there is a tag-effect caused by the FLAG
tag and the expression of SUSAA2-FLAG causes a new phenotype? Also, it is not clear to me
what promotor was used for the SUSAA2-FLAG constructs.

Sorry for the confusion. The previous picture was taken using T0 transgenic plants transplanted
from selection plates, which probably mislead the reviewer as the plants exhibit a different look
than normal (it is not a new phenotype). The picture is now replaced with a new one with
independent T2 lines grown directly on soil (without transplanting from plates) under the same
conditions as the controls (Fig S10). All three mutant versions of SUSAA2-FLAG, which are
driven by CaMV 35S promoter, rescued *saul1-1 susa2-1* mutants back to *saul1-1* like. The
functionality of WT-SUSAA2-FLAG control is shown by its ability to fully complement the
*saul1-1 susa2-1* phenotype back to *saul1-1* phenotype (Figure S2A).

We isolated *susa2-1* single mutant. As shown in Figure 4, all three *susa2* alleles show a WT-like
morphology and display slight susceptibility to *P.s.m.* ES4326. SUSAA2-FLAG transgenic lines in
WT background have a WT-like morphology and do not generate a new phenotype, the data of
which is not included in the manuscript.

Lines 244-246: Please provide images showing the non-complementation of the ACTIN domain

swapping construct in order to compare them to the other complementation. In addition, it would
be great to see western blots showing expression of used constructs for any experiment presented
(transgenic, complemented plant lines and transient expressed proteins in the split-luciferase
assay.).

As suggested, the figure of non-complementation of ACTIN domain swapping construct has
been added (Figure S11B-C).

In addition, gene expression data of used constructs in all the related experiments have been
added in Figure 2G, 3E, 4B, S2B, S4C, S10C, S11E, S12B, S15B and S16.

Lines 250-256: The authors write that two other *susa2* alleles (T-DNA alleles, *susa2-2* and
*susa2-3*) were isolated and characterized. However, beside the presented pathogen infection
phenotypes tested no image of the mutant plants in comparison to *susa2-1* are shown. Also, no
proof of the knockout of the said lines is provided (RT-PCR showing that these alleles are indeed
null alleles, especially since the two T-DNA lines used are depicted as intron insertions). Further
it would be nice to show that all three *susa2* alleles suppress *saul1-1* autoimmunity similar.

All mutant lines are morphologically WT-like (Figure 4A). When *SUSA2* expression was
examined by RT-PCR, *susa2-2* and *susa2-3* showed drastically reduced *SUSA2* expression while
*susa2-1* show similar *SUSA2* expression compared to Col-0 WT (Figure 4B).

Suppression of *saul1-1* by all three *susa2* alleles is now shown in Figure 2D. Their similar
suppression strengths indicate that they are all null alleles of *susa2*.

Figure 4: Why was the originally identified *susa2-1* allele not used side-by-side with the two T-
DNA alleles in this experiment?

Also, in Figure 4 the *eds1*, *ndr1*, *agb1* and *sid2* mutants were used as highly susceptible mutants
(controls) – it should be mentioned what these mutants are – at least in the figure legend or the
material and method part.

As suggested, *susa2-1* allele was isolated, the infection experiments were redone with all alleles,
and the new data are included in Fig.4.

For clarity, descriptions of *eds1*, *ndr1*, *agb1* and *sid2* control mutants have been added into the
figure legends as suggested.

Lines 272-273 and figure 5b: In the figure the PR1 expression in the *susa2-2* allele is also higher
than in the wt, was this a consistent phenotype? It would be nice if the authors could shortly
discuss this also in the manuscript. Further, it would be good to provide statistics for the results
in Figure 5b and c.

*PR1* expression level is highly sensitive to growth conditions, especially when the plants are
grown on soil. To address the reviewer's concern, *PR* genes expression was quantified this time
using plate-grown seedlings, the condition of which is more controllable. As shown in the new

Figure 5B-C, *susa2-2* allele shows similar *PR* gene expression compared to WT. Statistical
analysis was added in Figure 5B-C as suggested.

Figure 5A and E: I am not sure whether this is the normal variability of the *chs1-2* mutant
phenotype, but the *chs2-1* mutants in figure 5A and 5F look quite different, also they were grown
under the same conditions, right. Is this the normal phenotypical variation?

The *chs1-2* mutant shows chilling sensitive cell death phenotype when moved from 21°C to low
temperature growth condition (16°C). The morphological phenotype of *chs1-2* varies depending
on the conditions of the growth chamber used, the age of the plants before switching and the
induction time length of chilling sensitive cell death phenotype under 16°C. Slight change in any
parameter can lead to drastic difference in the morphology. This is a known phenotypical
variation. Again, the plant size is a more accurate indication of the autoimmunity consequence.

Figure 5D: authors present Ion leakage data obtained similar to what is shown in figure 1B, but
the percentage presented here are super low compared to figure 1B. Can the authors comment on
that please?

As different numbers of punched leaf discs in 25ml deionized water were used in each ion
leakage assay, the percentage of the absolute conductivity varies accordingly. These differences
in the experiments are now added into the figure legend for clarity.

Lines 264-265: The authors write that, in light of their pathogen infection assays, SUSAA2 plays a
positive role in plant immunity against virulent pathogens. I would agree in terms of the one
tested (Psm ES4326), but this is the only one tested, thus I suggest to specifically mention this.

The specific mentioning is now added as suggested.

Lines 282-290: Authors describe a whole set of experiment(s) – generating transgenic plants
overexpressing 35S::GFP-SAUL1 in Col-0 and the *susa2-2* mutant background and discuss the
obtained data to make the point that SUSAA2 is also required for the SOC3-TN2 mediated
35S::SAUL1 autoimmune phenotype. However, no data is presented for this experiment(s). To
make such a point, it would be great to see the actual data, including images of the transgenic
plants and either western blots or RT-PCRs showing the expression of the transgene in the
specific backgrounds and phenotypes (i.e. no or only a weak expression in non-phenotypic plants
vs. high or detectable expression in phenotypic plants).

The requested data are now shown in Figure S12A-B. *SAUL1* expression measured by RT-PCR
was added.

Lines 313-315: Authors present an experiment where they wanted to check whether the *susa2-2*
mutant can suppress the EDS1-YFP-NLS auto-immune phenotype and come to the result that the
*susa2-2* mutant cannot suppress this phenotype – indicating that SUSAA2 acts upstream of EDS1.
Given the data obtained from the earlier mentioned experiments in the manuscript and from
Disch et al., 2016) this was to be expected, since the dependency of SOC3 (and the autoimmune
mutants that require SOC3 presence) are all EDS1 dependent.

As the reviewer stated, our data agrees with the expectation that SUSAA2 should act along EDS1.
But our data do reveal that SUSAA2 is genetically upstream of EDS1, supporting its role at the
sensor NLR level, rather than downstream at the EDS1 module. This genetic data is important to
support SUSAA2's biochemical role at the sensor level, together with SOC3 and the TNs (see new
model in Fig 7).

Line 321: Authors write here that the "SOC3-CHS1 NLR complex is constitutively activated in
saul1-1 background..." - Is it really clear whether these NLRs are really 'activated'? The
experiments done here and in the past (as far as i can remember) just demonstrate that their
presence is required for the phenotype. Can a SOC3 or CHS1 P-loop mutant complement their
knock-out, for example in the saul1-1 soc3 or saul1-1 chs1-2 mutants, respectively? I think in
terms of what we know thus far about activity of NLRs required for (auto-)immunity, we should
be more specific in this regard and not talk to specifically about their activation when this was
not tested.

Current understanding of plant NLRs suggests that P-loop is required for NLR activation. Loss
of P-loop activity often leads to failed NLR activation. To address the reviewer's question, we
made CHS1 P-loop l-o-f mutant construct 35S::*CHS1-K213A* (Figure S15). 35S::*CHS1-K213A*
failed to complement *saul1-1 chs1*, indicating that P-loop activity, or activation of CHS1 is
indeed required for *saul1-1* autoimmunity.

Lines 335-336: Is this interaction experiment really a proof showing that SGT1b for being part of
an SCF SUSAA2 E3 ligase complex? I think it just indicates it, but does not show it! The only
thing it shows is that SUSAA2 can interact with SGT1b in transient overexpression in *N.*
*benthamiana* in a split-luciferase assay.

We changed the conclusion wording accordingly to be more accurate.

Figure 6C: Why is the positive control used in figure 6B not included in figure 6C?

The luminescence of positive control is too high and it masks the relatively weak interaction
intensity shown between SGT1b and SUSAA2. The control was therefore not included in Figure
6C. We added this explanation in the legend to be clear.

Figure 6D (and E): authors show their Co-IP results where they 'show' interaction of DN-
SUSAA2-FLAG with CHS1-HA (6D) and TN2-FLAGTEVZZ with DN-SUSAA2-FLAG. In figure
6D there are also bands in the negative control IP lane for both proteins or at least at a similar
size. Could the authors please comment on that. How can they be sure that what they see here is
really an interaction? Also, where is CHS1 in the input samples in figure 6D and where is DN-
SUSAA2 in the input samples in figure 6E? Maybe authors could provide an image of the whole
membrane and an longer exposure to show their presence in the input.

In Figure 6D, the weak band with similar size as CHS1-HA in the negative control lane after IP
was likely due to a weak binding affinity of CHS1-HA to anti-FLAG beads. However, DN-
SUSAA2-FLAG could consistently pull down a lot more CHS1-HA compared to the negative

control, suggesting a real interaction between CHS1 and SUSAA2. This type of observation is
common in IP experiments. For example, in Zhang *et al.*, 2017 in Scientific Reports (Long
noncoding RNA LINC00305 promotes inflammation by activating the AHRR-NF- κ B pathway in
human monocytes), in Figure 5D, enriched prey LIMR compared with the negative control
suggests interaction between the bait AHRR and the prey LIMR.

Even with long exposure time, we could not detect the input protein bands for CHS1-HA and
DN-SUSAA2-FLAG due to their low levels.

TurboID affinity labeling is a recently developed unbiased method to detect protein-protein
interactions, solving problems of detecting interactions including low affinity, low abundance, or
transience. To corroborate our biochemistry data, we additionally performed TurboID affinity
labeling experiment and co-IP assays in stable Arabidopsis transgenic lines to independently
confirm this important interaction. As shown in Figure 6E, immunoprecipitated DN-SUSAA2-
FLAG proteins could pull down CHS1-TurboHA and simultaneously be biotinylated by CHS1-
Turbo-HA. Furthermore, SUSAA2-FLAG could pull down CHS1-HA and TN2-HA in stable
transgenic Arabidopsis co-IP assay (Figure 6F and 6H), further supporting that SUSAA2
associates with SOC3-CHS1 or SOC3-TN2 NLR protein complex.

Line 372: Authors write here that the assembly of the SCFSUSAA2 E3 ligase complex facilitated
by the NLRs SOC3 and either TN2 or CHS1 is resulting in constitutive ETI (effector-triggered
immunity). Is this really an active process of assembling and activation, or is this complex
anyways assembled and through (outside) disruption activated? Also, I would not write
'constitutive ETI', since there is no evidence of a pathogenic effector protein involved.

The reviewer is right that it would be difficult to differentiate the two scenarios experimentally.
We corrected the mentioning accordingly.

Line 375: The manuscript has no Figure 8.

Mistake corrected.

Lines 379-380: here the authors write: "The assembly of such complex is likely facilitated and
maintained by HSP90.3/SUSAA3 and SGT1b chaperons." This is confusing, since in line 369 the
authors write that the NLRs facilitate the assembly of this complex. Who is now responsible of
the complex formation? Please try to make this clear.

This section has been revised accordingly. Our model has also been substantially altered during
revision. In our new model (Figure 7), both chaperons and the F-box protein are required for the
SCF assembly.

Lines 378-380: The conclusions the authors make here are not supported by their data – at least
in my view. Their hypothesis implies that this SCFSUSAA2 E3 ligase complex is not formed in
plants/cells that are not undergoing an immune response (or where SOC3/TN2/CHS1 NLRs are
not activated), right? To confirm such a hypothesis the authors should perform Co-IP
experiments with double transgenic Arabidopsis plants expressing functional tagged proteins for

example in wildtype and *saul1-1* background - here they should be able to see complex
formation only in the *saul1-1* mutant and not in wildtype. Otherwise the presented data does not,
in my view, support such a conclusion/hypothesis or comparison with the animal NLR
inflammasome recruiting caspsase-1.

To test this, Arabidopsis stable transgenic lines carrying SUSAA2-FLAG and CHS1-HA or TN2-
HA were generated. As shown in Figure 6F and 6H, SUSAA2-FLAG could pull down CHS1-HA
or TN2-HA in Arabidopsis co-IP assay in Col-0 WT background, supporting that SUSAA2
associates with SOC3-CHS1 or SOC3-TN2 protein complex in Col-0 WT plants. We were not
able to carry out IP experiments in the *saul1* mutant background as the plants are almost lethal.
As NLR activation is required for the *saul1* phenotypes (P-loop mutagenesis data in Fig S15), the
NLRs likely are activated in *saul1-1* plants undergoing constitutive immune responses.

Lines 403-405: The authors write: "..., the ACTIN domain of SUSAA2 cannot function as ACT2
and unlikely serves a canonical ACTIN function." I do not agree that this interpretation of the
experiments provided is correct, also part of the experiment results is not presented and rather
only mentioned. The SWAP experiment only shows that replacing the ACTIN domain of SUSAA2
with that of ACTIN2 results in a non-functional SUSAA2 protein not having the ability to
complement the *susa2-2* mutant. A function of the SUSAA2 ACTIN domain in complementing a
*actin2* mutant was not done.

The data and figure of ACTIN domain swapping experiment is now added (Figure S11). As
suggested, to further confirm this, we generated a *SUSAA2-ACTIN* domain construct and
transformed it into the *act2* mutant *der1-1* to check for complementation. As shown in Figure
S11D, *SUSAA2-ACTIN* failed to complement *der1-1*, indicating that the SUSAA2 ACTIN domain is
functionally divergent from ACT2.

Lines 406-408: Where is the data showing that SUSAA2 gene 'appearance' coincides with the
appearance of NLR genes in plant lineages? The 'evolutionary analysis in figure S4 and S5 do not
provide this data.

This sentence was removed to avoid confusion.

Figure S2: Authors provide an image of a Co-IP between HA-SAUL1 and DN-SUSAA2-FLAG
and see no interaction of DN-SUSAA2-FLAG with HA-SAUL1. Was this IP done in a reciprocal
manner as well with similar results? Could the C-terminal FLAG tag have an influence on this
interaction? NOTE that the complementation of the *susa2-2 saul1-1* by the FLAG-tagged SUSAA2
was not complete as well – at least in my point of view.

As suggested by the reviewer, we carried out the reciprocal IP, the data of which can be found in
Figure S3B. Consistently, SAUL1-C29A-FLAG could not pull down SUSAA2-HA.

FLAG-tagged SUSAA2 can fully complement *susa2-1 saul1-1* plants, as shown in Figure S2A.

Figure S3: here is also the positive control value missing in panel B.

The luminescence of the positive control is too high that it would mask the relatively weak
interaction intensity between SGT1b and SUSAA2. The control was therefore not included here.
We added the explanation in the figure legend to be clear.

Figure S6: It would be great if the authors could also include the sites of the introduced point
mutations that are supposed to affect canonical ACTIN function.

The requested information is now added to Figure S5 and S7.

Figure S8: Could the authors indicate the residues important for nucleotide binding, as
mentioned in the text and provide the PDB number of the structure used to model SUSAA2 actin
domain.

The residues important for nucleotide binding have been added in Figure S9A. Since the crystal
structure of SUSAA2 ACTIN domain has not been solved and there is no PDB number available,
Phyre2 webtool was used to predict the structural model of SUSAA2 ACTIN domain by using
SUSAA2 ACTIN protein sequence as input.

Reviewer #2 (Remarks to the Author):

The plant immune system relies on intracellular NLR receptors to directly and indirectly detect
pathogen virulence effectors. Exactly how these receptors function remains obscure. Xin Li's
group has expanded the number and diversity of NLR mechanisms by screening for suppressors
of autoactive NLRs. This has led to some interesting observations where full-length TIR-NBS-
LRR (TNL) receptors require "truncated" partner TIR-NB (TN) genes. The exact mechanism by
which hetero-NLR receptors function is unclear. In this work, the authors report on suppressor
mutants that block saul1-1 autoactive cell death. SAUL1 is an E3 ligase that appears to be
guarded by a combination of the TNL SOC3 and genomically neighboring TN genes CHS1 and
TN2. The hypothetical pathogen effector that triggers the SOC3/TN NLR system(s) remains
unknown. The suppressors encode a previously isolated mutation in HSP90.3 and a new F-box
protein SUSAA2. The repeated recovery of specific HSP90 alleles is an
interesting observation, and it's still unclear how these alleles differentially affect various NLRs.
The SUSAA2/ARF8 is an interesting suppressor with a fascinating domain structure. The
mechanism is unclear and not really tested here, but hopefully there will be follow-up stories.

I have no major reservations about this paper and think it can be published in essentially its
current form with edits to the text. I think that discussing HSP90 S100F in more detail (see
comments below) would improve the paper by more explicitly tying it back to the literature. The
text discussing the model figure and the figure itself could be more explicit about the supporting
data and its limitations.

Specific comments:

Line 138) Need to introduce the F-box motif here to help the reader.

Introduction of the motif is added as requested.

Line 138) Add references for ARP8
Check text for missing articles: e.g. Line 139) “A transgene complementation...”, Line 141)
“When the genomic region...”, Line 143) “..displayed a saul1-like phenotype”.
Missing articles have been added.
Line 153) List AA deleted in dnSUSA2 construct.
Details added as suggested.
Line 177) S100F in HSP90.3 same position as S100F allele of HSP90.2 previously found to
destabilize RPM1 (Hubert 2003). HSP90.2 and HSP90.3 are 99% identical. You should
comment more on the unusual recessive GOF phenotype of HSP90s in regards to destabilizing
NLRs. If you have tested an HSP90.3 KO mutant, it would be expected to not suppress? Also
worth testing for loss of RPM1 function in hsp90.3 S100F (Ah, I now see this is all tested in your
Huang et al MUSE paper as you independently isolated S100F again!?) You should include
SUSA3 in the discussion of this paper to comment on this more explicitly, currently there is
nothing?
More discussion is now added on this particular amino acid and its implications on its function.
As *susa3-1* is recessive in regards to *saul1* suppression (Fig 3A) and WT-*HSP90* can
complement the *susa3-1* defects (Fig 3D), we do not believe this allele is GOF here in regards to
saul1 suppression. We do not know why this specific mutation can behave differently in
different backgrounds. It could be its altered roles in different protein complexes.
We did cross a T-DNA insertional allele of *hsp90.3* (likely null) with *saul1* to try generating the
double mutant. However, after genotyping many plants, we were unable to find the double in F2.
In F3 progeny from F2 lines that were homozygous for one, but heterozygous for the other locus,
we were still unable to identify the double mutant, suggesting that the double is likely lethal. As
the T-DNA allele is already known to behave quite differently with the *hsp90.3-1* allele in the
literature, the *hsp90.3-1* allele we identified again is likely a unique partial LOF allele affecting
only certain clients of the chaperon. As this is not directly related to our NLR-SCF story, we did
not include this data to avoid confusion.
Line 181) Worth mentioning that previously isolated hsp90.3-1 is the same mutation (S100F) as
susa3-1.
The mentioning is added for clarity.
Line 237) “Therefore the ACTIN domain does not serve a canonical ACTIN function” This
seems over-interpreted. We don’t know that the hypothetical ACTIN function is required to
suppress saul1-1. It’s possible that SUSA2 protein accumulation is sufficient for saul1-1
suppression rather than SUSA2 function. What SUSA2 does in other contexts is unknown and
not assayed. Seems safer to just say that these residues are not required for SUSA2 function to

suppress saul1-1.

The claim is toned down as suggested.

Line 327-345; Figure7) Seems like the model is extremely hypothetical. Can be presented more
clearly by denoting pairwise interactions/evidence? Data presented are mostly pairwise coIP, but
this is not strong evidence for a multiprotein complex (beyond pairwise interactions)? So if A
coIPs B and B coIPs C, that is not sufficient information to say if there is an A/B/C complex, or
if there are independent A/B and B/C complexes.

The reviewer is right that our old model is too hypothetical. The model has been substantially
revised (Figure 7) combining all reviewers' suggestions.

Discussion: More discussion of a possible non-degradation/proteasome function for
ubiquitination by SUSA2/ARP8? ARP8 previously localized to nucleolus (Kandasamy, 2008), is
nucleolar function consistent with what is known about SOC/CHS1/TN2 localizations?

SOC3 was previously shown to localize to the nucleus, which is not in conflict with the
subcellular localization of SUSA2. The possibility of non-degradation/proteasome function for
ubiquitination by SUSA2/ARP8 is now added to the Discussion.

Figure 7: Don't call this a resistosome. Resistosome implies a ZAR1-like oligomer, no data
presented to support this?

Agree, the usage of the term resistosome for SOC3 is taken out from the manuscript.

Figure 7: Flip position of SOC3 and TIR-NBs as no coIP evidence for TIR-NBs and HSP/SGT?
E3 oval too large, just outline E3 components?

The figure is changed accordingly.

Reviewer #3 (Remarks to the Author):

The authors previously showed that loss of function of the PUB44 E3 ligase (SAUL1) leads to
activation of TIR-NLR pair SOC3-CHS1, whereas overexpression of PUB44 leads to activation
of TIR-NLR pair SOC3-TN2, suggesting that SOC3 pairs with CHS1 or TN2 to monitor PUB44
homeostasis. The authors hypothesize that the aforementioned TIR-NLR pairs monitor a
substrate (substrates) of PUB44. This study aims to identify this substrate by genetic screen. The
authors successfully identified two genes whose mutations suppressed autoimmune phenotype in
the pub44 background in which SOC3 is known to be activated. One gene encodes HSP90.3, the
other, called SUSA2, encodes an F-box protein with a C terminal extension sharing modest
homology with actin. Further genetic analyses confirmed that SUSA2 is required for both SOC3-
CHS1 and SOC3-TN2 function, but not other NLRs, indicating that SUSA2 is specifically
required for SOC3 function. Genetic analysis also suggested that SUSA2 acts upstream of
EDS1. The authors further show that SUSA2 can interact with TN2, CHS1, ASK1, and SGT1,
and HSP90.3 can interact with SOC3. The authors conclude that the aforementioned TIR-NLR

pairs and SUS A2 are assembled into a large SCF complex and that this process is facilitated by
HSP90.3 and SGT1. The authors also suggest that this complex allows the TIR-NLR pairs to
execute signaling by ubiquitinating and degrading negative regulators of defenses. While the
genetic analyses are sound and solid, the protein analyses are insufficient to support the
conclusions. Also, the proposed model is at odds with current understanding of TIR-NLRs.

Major comments:

1. Based on pair-wise protein-protein interaction experiments, the authors suggest that SOC3,
CHS1, HSP90, SGT1b, SUS A2 and ASK1 all exist in the same protein complex. It is at least
equally possible that they exist in different protein complexes. Can TN2 or CHS1 pull-down
ASK1 or CUL? Can ASK1 pull-down TN2, CHS1, or SOC3? Can these proteins be detected in
the same complex via gel filtration or blue native gel? Comprehensive analyses are needed
before such conclusion can be made.

As suggested, co-IP assay was performed to test ASK1-SOC3 interaction in Figure 6J. ASK1-
FLAGTEVZZ could pull down HA-FLAG-SCO3, supporting that ASK1 is in the SOC3
complex with CHS1 or TN2. ASK1-SUS A2 interaction in the SCF complex, as revealed by split-
luciferase assay and co-IP analysis (Figure S4), further supports the formation of SCF-NLR
complex. Many of the protein-protein interactions were also corroborated further with TurboID
(see response to reviewer 1, pg8).

Due to technical difficulties with SUS A2 and NLRs, all having extremely low protein abundance,
gel filtration or blue native gel experiments were not pursued.

2. Although PUB44 (SAUL1) interacts with the two TIR-NLR pairs, it does not interact with
SUS A2. This further argues against the idea that these proteins exist in the same complex. A
technical issue: A truncated SUS A2 protein lacking the F-box was used for protein-protein
interaction studies. Although this increases protein level, it is difficult to draw conclusion from a
lack of interaction between this construct and PUB44. If the F-box happens to be the interaction
domain, then the construct will not be able to detect the interaction.

SUS A2 contributes to the immunity mediated by two NLR pairs SOC3-CHS1 and SOC3-TN2,
which monitor their guardee SAUL1. Here SUS A2 likely serves as executor of defense.
Although both SAUL1 and SUS A2 interact with the NLR, they do not interact with each other
(see new model in Figure 7).

It is also possible they reside together in the same large NLR-SCF complex, the interaction of
which may not be detected by co-IP assay due to their physical distance. Such examples are not
uncommon in macromolecular complexes. For example, in SGT1/SCF/COP9 signalosome
complex which has roles in *N* gene-mediated resistance in tobacco, except for positive IP
interaction between SGT1 and CSN4, the interaction of SGT1 and other CSN subunits from
COP9 signalosome could not be detected using co-IP (Liu *et al.*, Plant cell, 2002). Another
example is that in the recently reported ZAR1 resistosome which includes decoy PBL2,
pseudokinase RKS1 and CNL ZAR1, PBL2-FLAG couldn't pull down ZAR1-HA in co-IP assay
(Wang *et al.*, Cell Host & Microbe, 2015).

For F-box proteins, the F-box domain has been shown previously to mediate specifically the
interaction with Skp1/ASKs. In the case of SUSAA2, the ACTIN-like domain should be the
putative interaction domain with the SCF ubiquitination substrate and other SCF assembly
components. Therefore, the F-box deletion SUSAA2 here serves as a dominant-negative (DN)
form of the protein which can stabilize the substrate rather than ubiquitinating it. Such DN
strategy has been reliably used to search for ubiquitination substrates of SCFs (Lee *et al.*, Plant
Physi., 2018). For example, circadian clock F-box E3 ligase ZEITLUPE (ZTL) ubiquitinates its
substrate CCA1 HIKING EXPEDITION (CHE), which was identified by utilizing DN-ZTL in
IP-MS to search for interactors.

3. The authors suggest that HSP90.3 and SGT1b facilitate the assembly of NLR-SCF complex.
What about the protein levels of SOC3, CHS1, and TN2? In addition to SCF complex assembly
in yeast and animals, HSP90 and SGT1b are known to be involved in stability control of NLRs
in both plants and animals. Also, it is necessary to show the SOC3-CHS1, SOC3-TN2,
CHS1/TN2-SUSAA2, and SUSAA2-ASK1 interactions are affected by mutations in HSP90.3 and
SGT1b. If HSP90.3 and SGT1b are required for the assembly of the NLR-SCF protein complex,
mutations of HSP90.3 and SGT1b are expected to impair the interactions.

As shown in Fig 6H input control, stability of TN2 does not seem to be significantly affected by
*hsp90*. However, in Fig 6F input, CHS1 seems to accumulate more in the chaperon mutants
(which is opposite to the reviewer's prediction). This agrees with previous knowledge that
HSP90 and SGT1b can affect NLR protein levels, sometimes positively, sometimes negatively.
Although Wanwan has now obtained constructs for the complementing tagged SOC3, it will take
a long time to move it from the stable parent lines to *sgt1b/hsp90* backgrounds. As this is not a
key point of the manuscript (no matter how the chaperons affect the NLR levels, it will not
change the conclusion that they are involved in the NLR-SCF assembly), I hope the reviewer
would agree that investigating the exact role of SGT1 and HSP90 to the NLR stability is out of
the scope of the current study.

As suggested, Arabidopsis stable transgenic lines carrying SUSAA2-FLAG and CHS1-HA or
TN2-HA were generated to test the NLR-SCF complexing. We did not test all possible
interaction combinations as it has been already well established that the chaperons are involved
in individual NLR or SCF complex assembly. As no one else is working on the SUSAA2 project,
Wanwan had to prioritize her time experiments and not overstretch herself.

As shown in Figure 6F and 6H, SUSAA2-FLAG could pull down CHS1-HA and TN2-HA in
Arabidopsis co-IP assay in Col-0 WT background, further supporting the formation of NLR-SCF
complex. However, the ability of SUSAA2-FLAG to pull down CHS1 or TN2 was greatly reduced
in *hsp90.3-1* and *sgt1b* mutant backgrounds, supporting the contribution of the chaperons to the
assembly of the NLR-SCF complexes.

4. The study started with PUB44, but their final model does not have a place for PUB44. Do
PUB44 and SUSAA2 affect each other in protein stability? Could PUB44 be a substrate for
SUSAA2?

SUSAA2 and SAUL1 doesn't affect each other's protein level when co-expressed in *N.*
*benthamiana* or *Arabidopsis* (new data provided in Figure S3C-E). Therefore, SAUL1 is unlikely
to be the substrate of SUSAA2, and vice versa.

We have substantially revised our working model. In the current one, SAUL1 serves as a
guardee and SUSAA2 serves as part of the SCF executor for NLR SOC3.

5. Additional evidence is needed to support the idea that an intact SCF complex is required for
the function of two TIR-NLR pairs.

It is known from the literature that F-box proteins are responsible for assembling SCF complex
through the interaction between the F-box domain and ASK/Skp1 proteins. In the case of
SCF^{SUSAA2}, SUSAA2 is the F-box protein which interacts with ASK1 to assemble the SCF complex.
The requirement of the SCF for the functions of the two TNL-TN pairs is supported by the
following data:

- 1. Loss of function of any of the SCF components, i.e., SGT1, HSP90, or SUSAA2, leads to
suppression of *saull1*, which is caused by the activation of SOC3-CHS1 pair. These
mutations all lead to SCF malfunction.
- 2. DN-SUSAA2- FLAG, in which the F-box was deleted, show failed complementation
(Figure S2B), suggesting that the assembly of the SCF through the F-box is required for
SOC3-CHS1 activation.

6. Even if a SCF complex is involved, it is problematic to envision the SCF complex as an
executor of immune signaling. Recent studies show that the TIR domain acts as a NADase which
presumably release an unknown product as a signal to trigger the EDS1-dependent immunity,
which is executed by NRG1 and ADR1 proteins. The idea that SUSAA2-mediated ubiquitination
and degradation of a negative regulator of defense does not explain the previous understanding
of TIR-NLR signaling. Also, SUSAA2 is only required for SOC3 but not other NLR function.
Most likely interpretation would be that SUSAA2 is monitored by SOC3-CHS1/TN2 pairs.

As no interaction can be detected between SAUL1 and SUSAA2, we do not believe SUSAA2 is part
of the guardee SAUL1 for the NLR pairs.

The reviewer raised an important point; we did not include recent important conceptual advance
on TNLs for our previous model. We have revised it substantially (Figure 7). We believe there
could be two scenarios to explain the NLR activation mechanism through the TNL-TN pairs here:

- 1. One product of TIR-NADase serves a molecular glue function for the SCF assembly.
This would be similar to the SCF^{TIR1} or SCF^{COI1} complexes, where the plant hormones
auxin and JA, respectively, serves as glue for the SCF (scenarios A and B).
- 2. The NLR-SCF complex serves dual functions, one is to oligomerize the TIR domain to
enable NADase activity, and simultaneously assembles the SCF for substrate
ubiquitination. Such dual function is not in conflict with the NADase model as the
NADase activity only explains the cell death phenotype caused by TNL activation.

7. In all split-luc experiments, empty vectors were used as negative controls. nLUC vector does
not contain a start codon, and it provides a false negative. This must be corrected.

As suggested, we replaced empty vectors with EDR4-CLuc and SNIPER8-NLuc as controls (Figure 6B and Figure S4A). The negative controls showed very weak or no signal.

Minor comments:

1. Lines 216-218: “The mutation site G252 in the ACTIN domain found in *susa2-1* mutant is also highly conserved, explaining its loss-of-function when mutated”. Any suggestion why this site affects function? Other conserved residues involved in nucleotide binding are not required.

The mutation site G252 might be involved in the interaction with the ubiquitination substrate of SUSAS2, thus showing suppression of *saul1-1* when mutated. This explanation is added in the text for clarity.

2. Does SUSAS2 play a role in PTI? The authors measured bacterial growth for *hrcC* mutant. It is necessary to thoroughly measure PTI responses.

As suggested, MPK activation was measured in *susa2* mutant alleles. As shown in Figure 4G, all *susa2* mutant alleles show WT-like *flg22*-induced MPK activation.

3. In discussion, the comparison with caspase in inflammasome may not be appropriate. Again, there is no evidence that SUSAS2 is an executor. The fact that it is specifically required for SOC3 but not other NLR function suggests that it is likely a guardee, not an executor. What is the phenotype of SUSAS2 overexpression plants?

Due to the large reservoir of F-box protein encoding genes in higher plants, we hypothesize that the SCF type of NLR complex assembly can happen with specific NLRs and dedicated F-box proteins, but common SGT1/HSP90. This explains the specific phenotypes of *susa2*, but a general NLR defect of *sgt1* and *hsp90* mutants. In a recent report, maize FBL41 is involved in race-specific defence. However, whether it acts together with NLRs or not is not known. As *susa2* is the first F-box protein mutant identified to support such model, future identification of more F-box proteins in other TNL complexes would further enhance such hypothesis.

We did try to examine the SUSAS2 overexpression phenotype. Around 18 independent transgenic lines of *35S::SUSAS2* in Col-0 were obtained, which all show wildtype like morphology. Among all transgenic lines with *35S::SUSAS2-FLAG*, no detectable protein bands can be seen on western blots. We suspect that SUSAS2 protein levels are exquisitely controlled, and cannot be overexpressed to avoid autoimmunity (its overexpression likely leads to lethality). As this is out of the scope of the current manuscript, we did not pursue this direction further.

4. Line 375, “Fig 8” should be “Fig 7”.

Corrected.

5. Fig S5, the analysis should include lower plants in which TIR-NLRs first appear.

The phylogenetic tree has been reconstructed as suggested.

REVIEWERS' COMMENTS

Reviewer #1 (Remarks to the Author):

Review of revision of manuscript NCOMMS-19-1125103A submitted to Nature communications entitled:

SCFSUSA2 E3 ligase is required for autoimmunity mediated by paired NLRs 2 SOC3-CHS1 or SOC3-TN2

by Wanwan Liang, Meixuezi Tong and Xin Li

In the revised version and also the comments to the reviewers, the authors deal with all important comments and criticisms in a (more than) satisfactory manner. The manuscript has substantially improved and adds important details to our current knowledge of NLR activation and potential downstream events. The authors also provide a new working model/hypothesis (Figure 7 and discussion) that could put their findings into a more general mode of NLR activation.

There are just a few minor issues:

lines 165-166: This sentence can be deleted, since it is redundant to what is written in lines 158 ff.

line 485: The name of first author of the reference cited here should be Li, N and not Li, X, right?

The presented and discussed model is a nice hypothesis and could indeed be true, but awaits the identification of further F-Box proteins required for NLR activation – as the authors mention. However, with different F-Box proteins being potentially identified most likely different ubiquitination targets will be identified as well and this raises a second question/problem: Why should there be so many different targets deployed/required or necessary for the initiation of NLR-triggered immunity? Would it not make more sense for the plant to use such a SCF-complex activation mode to degrade or 'activate' certain key negative or positive regulators of immunity, respectively? Or do the authors place this NLR-SCF-complex activation upstream of the conserved key regulators?

Reviewer #2 (Remarks to the Author):

In this revision, the authors have helpfully improved several aspects of the manuscript, but have also introduced new text and data that are less helpful. The model as presented, in Figure 7 has become even more specific, with events that are not addressed by the paper. At the least, the authors need to add "hypotheses"/"hypothetical models"/similar to the figure title/legend/discussion. Ultimately, the readers will decide what they are going to believe, but I think it would be better presented more explicitly as speculation. At the end of the paper, I'm not clear on why I should think that SUS2 is a specific NLR output adaptor rather than being specifically involved in SOC3 activation. The mutants are interesting though!

1) The manuscript still has not been proofread and contains an excessive number of grammatical errors (mostly involving missing articles, subject/verb agreement and spelling) that should be fixed before publication. I'll list several below, but they are just a start.

2) I think that the model is still extremely specific given the evidence. The evidence for a large complex is based solely on a pile of pair-wise interactions in many different assays (most of which are not measuring direct interactions). I think that the authors should make an explicit statement in the discussion that they have no direct evidence for a large complex where all these proteins are simultaneously found, nor data getting testing the hypothesis that the interactions are indicative of an "executor" complex.

3) Figure S11 has been added where the authors do some domain swapping and truncations between SUSAA2's Actin domains and the ACT2 protein. These experiments don't help clarify the role of the ACTIN domains and certainly don't say if SUSAA2's ACTIN domain is "canonical" or not. The failure of these chimeras to complement either *susa2* or *der1-1* is uninterpretable. These sorts of experiments are only interpretable if they actually complement. I'd just present the previous site-directed mutants and leave it at that.

As an analogy, you could easily do these sorts of swaps between much more closely related proteins, say NB domains of 2 NLRs, and end up with non-functional chimeras. This would not support the idea that the NB domain of the first NLR was "non-canonical".

Typos and comments:

108: "...a truncated..."

114: "...a suppressor..."

133: "...the Col-0..."

147: "A transgene..."

148: "...the genomic..."

159: "As WT SUSAA2-FLAG..."

160: delete "likely due to self-ubiquitination and degradation" unless there are some supporting data.

162: change to "likely abolishing..." unless there are some supporting data.

167: "...an approach..."

169: "...pull down..."

174: "...confirming..."

177: "...bring their ubiquitination substrate into proximity with an E2 ubiquitin conjugating enzyme..."

191: delete "ecotype"

192: delete "was"

206: Delete "This amino acid", change to "Serine 100 is located in the ATP-binding pocket of HSP90.3, a highly conserved region..."

210: "we crossed the previously reported *hsp90.3-1* mutant with..."

211: replace "morphology" with "auto-immune phenotype"

263: "alanines"

269: "This indicates that G258..."

274: "...a domain swapping experiment was performed..."

283: Figure S11 and associated text: remove? "Taken together, the ACTIN domain of SUSAA2 is unlikely serving a canonical ACTIN-like function." Unclear why 278-283 would be expected to work, doesn't really inform whether or not SUSAA2 ACT domain is canonical. In any case, this should be phrased as "SUSAA2's function in regards to *saul1-1*". Site-directed mutations in preceding paragraph are enough.

390: Split-luciferase positive does not indicate a direct interaction.

462: Perhaps use "analogous" instead of "remotely similar"?

Reviewer #3 (Remarks to the Author):

The authors have addressed some of the issues raised during first review. However, whether a SOC3-SUSA2-SCF complex is formed upon resistosome formation is still not supported by the data. All tested interactions occur in WT background when SOC3 is not activated, suggesting that the components exist in a preformed complex. Thus SUSA2-SCF complex might be required for the oligomerization of SOC3 upon perturbation of SAUL1 rather than execution of immune signaling. Also, the authors argue that PBL2 could not pull down indirect interactor ZAR1. This is not true. The UMP-modified PBL2 does pull down ZAR1 in Wang et al., 2015 paper. I suggest the authors further tone down their claim and discuss the possibility that the aforementioned complex may well be a preformed complex before NLR activation.

We sincerely thank the three expert reviewers for reviewing our manuscript, which helps us to improve the quality and readability of our story. Here, our response details can be found in blue font underneath each original comment.

REVIEWERS' COMMENTS

Reviewer #1 (Remarks to the Author):

Review of revision of manuscript NCOMMS-19-1125103A submitted to Nature communications entitled:

SCFSUSA2 E3 ligase is required for autoimmunity mediated by paired NLRs 2 SOC3-CHS1 or SOC3-TN2

by Wanwan Liang, Meixuezi Tong and Xin Li

In the revised version and also the comments to the reviewers, the authors deal with all important comments and criticisms in a (more than) satisfactory manner. The manuscript has substantially improved and adds important details to our current knowledge of NLR activation and potential downstream events. The authors also provide a new working model/hypothesis (Figure 7 and discussion) that could put their findings into a more general mode of NLR activation.

There are just a few minor issues:

lines 165-166: This sentence can be deleted, since it is redundant to what is written in lines 158 ff.

As suggested, the sentence has been deleted.

line 485: The name of first author of the reference cited here should be Li, N and not Li, X, right?

Mistake corrected.

The presented and discussed model is a nice hypothesis and could indeed be true, but awaits the identification of further F-Box proteins required for NLR activation – as the authors mention. However, with different F-Box proteins being potentially identified most likely different ubiquitination targets will be identified as well and this raises a second question/problem: Why should there be so many different targets deployed/required or necessary for the initiation of NLR-triggered immunity? Would it not make more sense for the plant to use such a SCF-complex activation mode to degrade or 'activate' certain key negative or positive regulators of immunity, respectively? Or do the authors place this NLR-SCF-complex activation upstream of the conserved key regulators?

The reviewer raised a very important point. We added some discussion to include such likelihood. It is possible that different F-box proteins are used for NLRs at the sensor level, but they all target the same master downstream regulator for ubiquitination and defense activation.

Reviewer #2 (Remarks to the Author):

In this revision, the authors have helpfully improved several aspects of the manuscript, but have also introduced new text and data that are less helpful. The model as presented, in Figure 7 has become even more specific, with events that are not addressed by the paper. At the least, the authors need to add “hypotheses”/“hypothetical models”/similar to the figure title/legend/discussion. Ultimately, the readers will decide what they are going to believe, but I think it would be better presented more explicitly as speculation. At the end of the paper, I’m not clear on why I should think that SUSAN2 is a specific NLR output adaptor rather than being specifically involved in SOC3 activation. The mutants are interesting though!

1) The manuscript still has not been proofread and contains an excessive number of grammatical errors (mostly involving missing articles, subject/verb agreement and spelling) that should be fixed before publication. I’ll list several below, but they are just a start.

Corrected as suggested below.

2) I think that the model is still extremely specific given the evidence. The evidence for a large complex is based solely on a pile of pair-wise interactions in many different assays (most of which are not measuring direct interactions). I think that the authors should make an explicit statement in the discussion that they have no direct evidence for a large complex where all these proteins are simultaneously found, nor data getting testing the hypothesis that the interactions are indicative of an “executor” complex.

The statement has been added into the discussion to be clear that we so far do not have evidence for a large complex formation based on pairwise protein-protein interaction analysis. .

3) Figure S11 has been added where the authors do some domain swapping and truncations between SUSAN2’s Actin domains and the ACT2 protein. These experiments don’t help clarify the role of the ACTIN domains and certainly don’t say if SUSAN2’s ACTIN domain is “canonical” or not. The failure of these chimeras to complement either susan2 or der1-1 is uninterpretable. These sorts of experiments are only interpretable if they actually complement. I’d just present the previous site-directed mutants and leave it at that.

As an analogy, you could easily do these sorts of swaps between much more closely related proteins, say NB domains of 2 NLRs, and end up with non-functional chimeras. This would not support the idea that the NB domain of the first NLR was “non-canonical”.

We agree with the reviewer and have removed Figure S11 as suggested.

Typos and comments:

108: "...a truncated..."
114: "...a suppressor..."
133: "...the Col-0..."
147: "A transgene..."
148: "...the genomic..."
159: "As WT SUS A2-FLAG..."
160: delete "likely due to self-ubiquitination and degradation" unless there are some supporting data.
162: change to "likely abolishing..." unless there are some supporting data.
167: "...an approach..."
169: "...pull down..."
174: "...confirming..."
177: "...bring their ubiquitination substrate into proximity with an E2 ubiquitin conjugating enzyme..."
191: delete "ecotype"
192: delete "was"
206: Delete "This amino acid", change to "Serine 100 is located in the ATP-binding pocket of HSP90.3, a highly conserved region..."
210: "we crossed the previously reported hsp90.3-1 mutant with..."
211: replace "morphology" with "auto-immune phenotype"
263: "alanines"
269: "This indicates that G258..."
274: "...a domain swapping experiment was performed..."

All Corrected.

283: Figure S11 and associated text: remove? "Taken together, the ACTIN domain of SUS A2 is unlikely serving a canonical ACTIN-like function." Unclear why 278-283 would be expected to work, doesn't really inform whether or not SUS A2 ACT domain is canonical. In any case, this should be phrased as "SUS A2's function in regards to saul1-1". Site-directed mutations in preceding paragraph are enough.

We agree with the reviewer and have removed Figure S11 as suggested.

390: Split-luciferase positive does not indicate a direct interaction.

As suggested, "directly" was removed.

462: Perhaps use "analogous" instead of "remotely similar"?

Changed as suggested.

Reviewer #3 (Remarks to the Author):

The authors have addressed some of the issues raised during first review. However, whether a SOC3-SUSA2-SCF complex is formed upon resistosome formation is still not supported by the data. All tested interactions occur in WT background when SOC3 is not activated, suggesting that the components exist in a preformed complex. Thus SUSA2-SCF complex might be required for the oligomerization of SOC3 upon perturbation of SAUL1 rather than execution of immune signaling. Also, the authors argue that PBL2 could not pull down indirect interactor ZAR1. This is not true. The UMP-modified PBL2 does pull down ZAR1 in Wang et al., 2015 paper. I suggest the authors further tone down their claim and discuss the possibility that the aforementioned complex may well be a preformed complex before NLR activation.

We have toned down our claim and added some discussion as suggested.